# An Algorithm for Deriving the Topology of Below-ground Urban Stormwater Networks

Taher Chegini[1] and Hong-Yi Li[1]

[1]University of Houston, 4226 Martin Luther King Boulevard, Houston, TX 77204, USA

**Correspondence:** Hong-Yi Li (hli57@uh.edu)

**Abstract.** Below-ground Urban Stormwater Networks (BUSNs) are critical for removing excess rainfall from impervious urban areas and preventing or mitigating urban flooding. However, available BUSN data are sparse, preventing modeling and analyzing urban hydrologic processes at regional and larger scales. We propose a novel algorithm for estimating BUSNs by drawing on the concepts from Graph theory and existing, extensively available land surface data such as street network, topography, and land use/land cover. First, we derive the causal relationships between the topology of BUSNs and urban surface features based on the graph theory concepts. Then, we apply the causal relationships and estimate BUSNs using web-service data retrieval, spatial analysis, and high-performance computing techniques. Finally, we validate the derived BUSNs in the metropolitan areas of Los Angeles, Seattle, Houston, and Baltimore in the U.S., where real BUSN data are partly available to the public. Results show that our algorithm can effectively capture 59-76% of the topology of real BUSN data, depending on the supporting data quality. This algorithm has promising potential to support large-scale urban hydrologic modeling and future urban drainage system planning.

## 1 Introduction

Urban flooding events pose escalating threats to urban areas at the regional and larger scales. The worsening of the urban flooding issue can first be attributed to urbanization and associated regional population migration. The United Nations estimates that globally the urban population will grow to more than two-thirds of the total population by 2050 (UN, 2019). This worsening can also be due to climate change, particularly accelerated extreme precipitation and sea-level rise, which have imposed increasing threats to urban populations (Schreider et al., 2000; Yang et al., 2013; Hettiarachchi et al., 2018; Rosenberger et al., 2021). Both urbanization and climate change are widely considered large-scale phenomena and have been studied mostly at the regional and larger scales (e.g., Ajaaj et al., 2017; Ntelekos et al., 2010; Teuling et al., 2019; Pang et al., 2022; Qian et al., 2022). As such, it is necessary to understand, predict, and mitigate urban flooding at regional and larger scales.

Generally, there are two types of systems for transporting stormwater from urban areas to local water bodies: Combined sewer system (CSS) and separate sewer system (SSS). CSSs collect domestic sewage and/or industrial wastewater in addition to stormwater whereas SSSs have two separate systems for collecting stormwater and sewage/wastewater. During heavy rainfalls, the overflows from CSSs are a major source of pollution, therefore, at the end of the 20th century, many major countries around the globe started adopting SSSs in their urban development plans and partially or fully transforming their existing CSSs into

SSSs (Mannina and Viviani, 2009). Although there are still many cities around the world with CSSs, SSSs are more common in many major countries. For example, there are over 700 communities in the U.S. that use CSSs while over 80% of the U.S. population resides in areas with SSSs (EPA, 2018). In China, the percentage of SSS and CSS usage varies by region, but overall, SSSs are predominant by accounting for 58%-87% of the total sewer line length in different regions of China (Huang et al., 2018).

The U.S. Environmental Protection Agency (EPA) uses the term Municipal Separate Storm Sewer System (MS4) to refer to the stormwater collection part of SSSs. In this study, we focus on MS4s since they are the most dominant type of stormwater transport systems in the U.S. EPA defines an MS4 as a publicly-owned urban stormwater conveyance system that directly receives excess surface runoff from urban areas during storm events and delivers it to lakes, rivers, or oceans. Thus, MS4s play an irreplaceable role in preventing or mitigating urban floods. However, due to a lack of good quality MS4 data, most urban modules in existing hydrological models focus on surface hydrological processes (e.g., those associated with impervious areas and compacted soils) and do not explicitly account for MS4 (e.g., Rafee et al., 2019; Qian et al., 2022; Cuo et al., 2008; Yang et al., 2011). One of the few exceptions is the Storm Water Management Model (SWMM) model (Rossman and Simon, 2022). SWMM nevertheless can only apply to a local, small-city level due to its intensive computational demand and requirement of detailed MS4 data, which typically are not available to the public (e.g., Nanía et al., 2015; Meyers et al., 2021; Fraga et al., 2016; Naves et al., 2019; Yang et al., 2011). Moreover, some studies instead of using the actual urban drainage network, generate synthetic networks based on probabilistic methods that can capture the hydrologic responses of urban watersheds (e.g., Seo and Schmidt, 2014; Kim et al., 2021). Generating such synthetic networks require parameter calibration based on hydrological observations such as streamflow. Therefore, MS4 data sparsity remains a grand challenge in urban hydrology, preventing us from understanding and modeling below-ground urban hydrologic processes at the regional or larger scales that are compatible with the impacts of urbanization and climate change.

In this study, we focus on the Below-ground Urban Stormwater Network (BUSN) elements of MS4s, i.e., above-ground elements such as street inlets, manholes, and ditches are not the subject of this study. We attempt to address the data scarcity challenge based on two premises. The first premise is the topological relationship between street/road networks and BUSNs. Generally, BUSNs are required to protect streets/roads from flooding and thus often constructed parallel to street/road networks, particularly for those important streets, as illustrated in Figure 1. The Urban Drainage Design Manual by the U.S. federal government (Brown et al., 2013) states that the main design objective of BUSNs is to collect the stormwater runoff and convey it along and through streets toward suitable water bodies without adversely impacting the streets' intended functions. The second premise is that above-ground urban geospatial data are now extensively available. For example, OpenStreetMap (OpenStreetMap, 2021) provides a vector-format, global map of street/road networks freely accessible to the public. By capturing and utilizing this topological relationship, we can derive the topological properties of BUSNs (e.g., geographic locations of stormwater pipes and their spatial connections) based on the existing street/road network and other above-ground data.

Our primary objective is thus to propose, develop, and validate a novel algorithm for deriving BUSN topological properties from ubiquitous existing above-ground data. The rest of the article is structured as follows. Section 2 describes the conceptual basis and technical details of the new algorithm. Section 3 lists four metropolitan areas in the U.S. as the case studies. Section

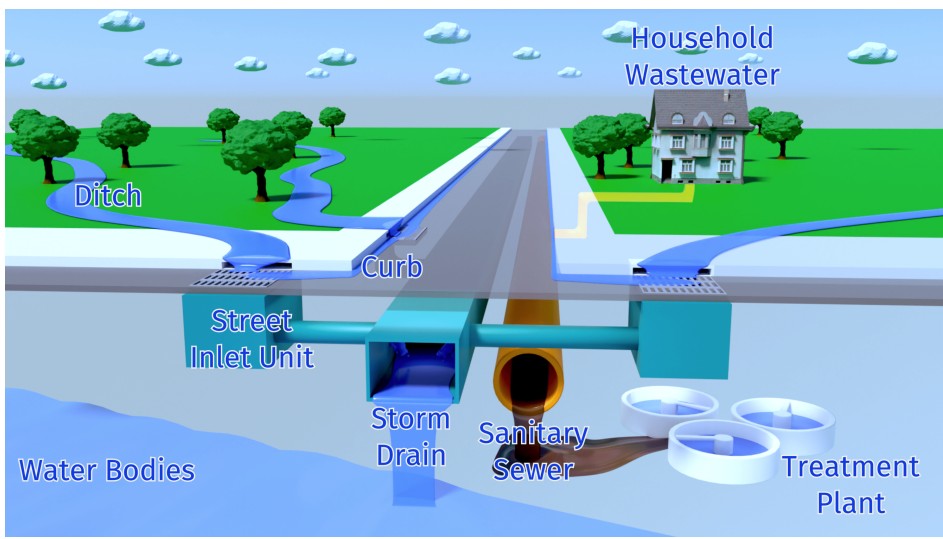

**Figure 1.** A schematic of an urban drainage system adapted from Town of Gilbert, AZ (2022).

4 illustrates this algorithm in these case studies. Section 5 closes with a summary and discussions on the limitations and future implications.

## 2 Methodology

In this section, we first explain the conceptual framework of our algorithm for deriving the topology of BUSNs, including 1) complex network analysis concepts from Graph theory that we adapt to capture the topological relationships between BUSNs and street/road networks. These concepts are transferable and not specific to any location. 2) A generic procedure for BUSN derivation. 3) A simple yet effective way for validating the derived BUSNs by the algorithm. 4) The technical details in implementing the algorithm in the U.S. are based on the federal and local urban drainage design criteria in the U.S. and publicly available land surface data.

### 2.1 Generic and Conceptual Framework

#### 2.1.1 Complex Network Analysis

BUSNs and street networks are both complex, hierarchical networks. The function of such networks largely depends on their nontrivial topological structure (Strogatz, 2001). Therefore, analyzing complex networks is usually carried out by measuring their structural properties. Graph theory introduces different types of structural properties for quantifying the function of complex networks from different perspectives such as clustering (grouping elements based on their attributes), connectivity (resilience to removing elements), and centrality (relative importance of elements). Since our proposed algorithm is based on the relative importance of elements of a complex network, centrality-based metrics are suitable for our algorithm. Moreover,

since in this study we are interested in analyzing flow paths in street and storm drain networks, we opt for using Betweenness Centrality (BC, Freeman (1977)) as a metric for measuring the relative importance of edges in a complex network such as BUSN.

The mathematical definition of BC is as follows:

$$\text{BC}(i) = \sum_{v \neq i \neq w} \frac{\sigma_{vw}(i)}{\sigma_{vw}}, \tag{1}$$

where $\sigma_{vw}$ is the total number of the shortest paths from Node $v$ to Node $w$, and $\sigma_{vw}(i)$ is the subset of paths that pass through Node $i$ (Brandes, 2001). In other words, the BC value of an edge is the ratio of the total number of the shortest paths that pass through the edge to the total number of the shortest paths in the entire network. In general, the shortest path between two nodes in a network is the path with the minimum number of edges that connect the two nodes. The specific definition of the shortest path closely depends on the network type. A network can be either undirected or directed. In an undirected network, there is a symmetric relationship between a pair of connected nodes, whereas, in a directed network, there could be a one-way relationship between a pair of connected nodes. For example, pedestrian pathways in a street network can be considered as an undirected graph since pedestrians can cross any path in the network in both ways. Car pathways, on the other hand, cannot be considered undirected since there are one-way and two-way streets. Therefore, in this case, the street network can be considered as a directed graph. A BUSN, nonetheless, is a directed network because the pipes in a BUSN are always one-way by design, i.e., stormwater moves in the pipes under gravity towards surface water bodies. In addition to direction, network edges can have numeric properties, such as length and width. These numeric properties, which are called edge weights, can be used to differentiate the importance or capacity of the edges. Within a directed network, BC can be calculated with or without the edge weights and denoted as weighted or unweighted BC, respectively.

We demonstrate the difference between weighted and unweighted BC in a directed graph through a simple example. Figure 2a shows an unweighted directed network in which for simplicity we assume the network is uniformly weighted, i.e., all edge weights are 1. Figure 2c is the same graph, but two edges have different weights, i.e., Edges $d \rightarrow f$ and $e \rightarrow f$ have weights of 9 and 3, respectively.

Table 1 shows all the paths in the network and the sum of edge weights along the paths. In the table, $\sum W_{ij}^{u}$ and $\sum W_{ij}^{w}$ are the sums of edge weights in a path for the uniformly weighted and weighted graphs, respectively. We note that since in the uniformly weighted graph all edge weights are 1, $\sum W_{ij}^{u}$ becomes the number of edges in a path.

The paths highlighted with a box in Table 1 are the shortest paths unique to each graph. For example, there are two paths between Nodes $a$ and $f$, namely $a \rightarrow d \rightarrow f$ and $a \rightarrow d \rightarrow e \rightarrow f$. In the uniformly weighted graph, the sum of weights for these two paths are 2 and 3, respectively, thus $a \rightarrow d \rightarrow f$ is the shortest path. On the other hand, in the weighted graph, the sum of weights for these two paths becomes 10 and 5, respectively, thus $a \rightarrow d \rightarrow e \rightarrow f$ becomes the shortest path from $a$ to $f$. By repeating this procedure for all the edges in the two networks, we can identify all the shortest paths passing through each edge and compute their BC values (Figures 2b and 2d). From this example, we conclude that edges with lower weights are more likely to have higher BC values since shortest paths are more likely to pass through them.

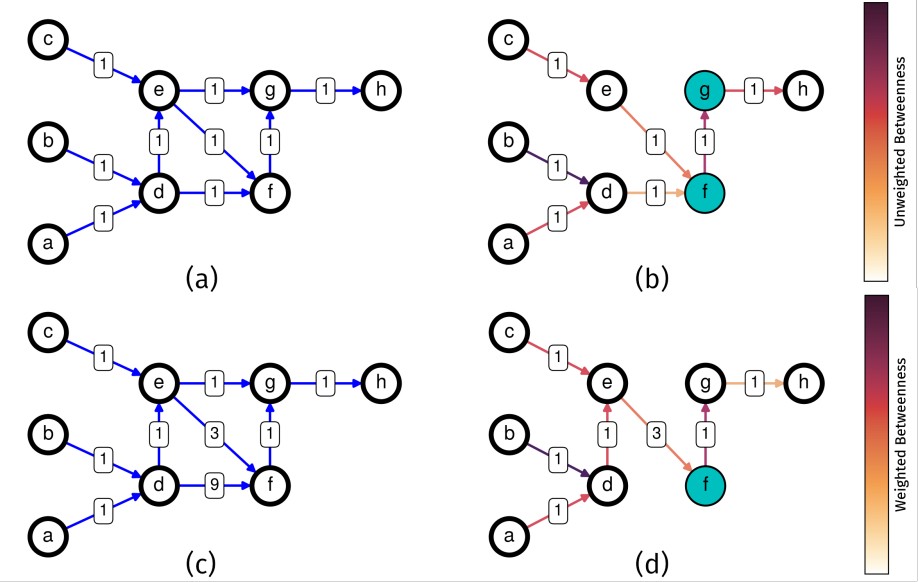

**Figure 2.** A comparison of unweighted ((a) and (b)) and weighted ((c) and (d)) betweenness centrality for two directed graphs. (a) and (c) show the edge weights. (b) and (d) illustrate the importance of edge weights in determining the shortest paths. The line colors in (b) and (d) represent the BC values.

Furthermore, the weights of edges can be calculated in various ways depending on their network function. Kirkley et al. (2018) adapted the BC concept in street network analysis to study traffic congestion, i.e., slower movement of vehicles due to the imbalance of street capacity and traffic volume. Considering a street network as a directed graph, street segments are edges and street junctions are nodes, and a higher weight associated with a street segment implies less importance of this street

segment. Since travel time is central to quantifying the congestion, they assigned street length as edge weight for computing BC. As such, for any street, a lower weight implies shorter travel time, a higher chance of this street being on an optimum traffic route, and thus higher importance. They showed that, in a street network, streets with high BC values are guaranteed to be a part of the backbone structure of the network and thus more important to traffic analysis. They nevertheless pointed out that using street length as the only edge weight was a limitation in their analysis since the length is not the only factor affecting

traffic congestion.

For adapting the BC concept to BUSNs, we rely on two facts: 1) A BUSN is also a complex network and a directed graph. 2) BUSNs are well-connected to street networks through some connecting elements such as street inlets and catch basins, as shown in Figure 1, as required by federal and state regulations (Brown et al., 2013). Since these connecting elements are primarily aligned with streets, we consider that below-ground stormwater pipes, for the most part, are laid beneath streets.

Therefore, we assume a BUSN topology is analogous to a street network's topology in an urban area, and we can infer the former from the latter. We also note that in a BUSN, pipes are edges and pipe junctions are nodes.

**Table 1.** The shortest paths in the two graphs shown in Figure 2

| $P_{ij}^u$ | $\sum W_{ij}^u$ | $P_{ij}^w$ | $\sum W_{ij}^w$ |
|---|---|---|---|
| $a \to d$ | 1 | $a \to d$ | 1 |
| $a \to d \to e$ | 2 | $a \to d \to e$ | 2 |
| $a \to d \to e \to g$ | 3 | $\boxed{a \to d \to e \to f}$ | 5 |
| $a \to d \to e \to g \to h$ | 4 | $a \to d \to e \to g$ | 3 |
| $\boxed{a \to d \to f}$ | 2 | $a \to d \to e \to g \to h$ | 4 |
| $b \to d$ | 1 | $b \to d$ | 1 |
| $b \to d \to e$ | 2 | $b \to d \to e$ | 2 |
| $b \to d \to e \to g$ | 3 | $\boxed{b \to d \to e \to f}$ | 5 |
| $b \to d \to e \to g \to h$ | 4 | $b \to d \to e \to g$ | 3 |
| $\boxed{b \to d \to f}$ | 2 | $b \to d \to e \to g \to h$ | 4 |
| $c \to e$ | 1 | $c \to e$ | 1 |
| $c \to e \to f$ | 2 | $c \to e \to f$ | 4 |
| $c \to e \to g$ | 2 | $c \to e \to g$ | 2 |
| $c \to e \to g \to h$ | 3 | $c \to e \to g \to h$ | 3 |
| $d \to e$ | 1 | $d \to e$ | 1 |
| $d \to e \to g$ | 2 | $\boxed{d \to e \to f}$ | 4 |
| $d \to e \to g \to h$ | 3 | $d \to e \to g$ | 2 |
| $\boxed{d \to f}$ | 1 | $d \to e \to g \to h$ | 3 |
| $e \to f$ | 1 | $e \to f$ | 3 |
| $e \to g$ | 1 | $e \to g$ | 1 |
| $e \to g \to h$ | 2 | $e \to g \to h$ | 2 |
| $f \to g$ | 1 | $f \to g$ | 1 |
| $f \to g \to h$ | 2 | $f \to g \to h$ | 2 |
| $g \to h$ | 1 | $g \to h$ | 1 |

Boxed paths indicate the paths that are in one graph and not in the other.

Moreover, it is neither necessary nor feasible to have a below-ground stormwater pipe below each street. For example, a country road or a street in a very sparsely populated area may not need a below-ground stormwater pipe since the corresponding surface infrastructures such as flood buffering zones may be sufficient to protect it from floods. Those streets in residential and business areas are relatively more important and will need BUSNs for flood protection due to two possible reasons: 1) The streets are so important that extra flood protections are needed besides those surface infrastructure (e.g., buffering zone, and retention ponds); 2) There is not enough space for surface infrastructure in heavily populated urban areas, where BUSNs is the most economical and feasible option. Indeed, within a street/road network, some streets are more important than

others depending on several factors, such as road types, urban form (e.g., street circulation system, buildings' arrangement, and distribution, and spatial accessibility), land use (such as residential, commercial, and industrial), and land cover (for example, open spaces, parks, and impervious surfaces). Those streets with more importance are thus more likely to have BUSNs underlying them. In this study, we quantify the relative importance of streets by incorporating the aforementioned factors into the BC concept.

To address the single-weight-factor limitation pointed out in the (Kirkley et al., 2018) study, for BUSN derivation, we propose to incorporate multiple stormwater-relevant factors, such as street topography and land use/land cover, into edge weights. In other words, instead of using only one attribute as an edge weight, we compute an integrated weight from multiple attributes. The resulting weighted BC reflects the integrated impacts of various urban factors on the required stormwater transport capacity of the BUSN pipe, hence denoted as integrated-weighted BC (IWBC) hereinafter. Intuitively, a street with a higher weight will have less requirement for stormwater transport due to having less importance and thus have a lower IWBC value. Correspondingly, this street will less likely have a BUSN pipe underlying it. In this study, we assign street weights based on the following rules:

- Since for computing BC, an edge should have a single integrated weight, different street/pipe attributes should be summarized into a single value.

- We transform the values of street/pipe attributes into edge weights such that streets/pipes with more significance have lower weights. The reason is that we measure the street/pipe significance based on their BC values and edges with lower weights are more likely to have higher BC values.

- Considering that street/pipe attributes can have different ranges or even data types, we normalize their values before assigning them as edge weights.

In this study, we consider four street attributes, namely, land cover type, road type, the discharge capacity of its associated storm drain pipe, and building footprint, with data types of integer, string, float, and float, respectively. First, we normalize each attribute by data binning, i.e., dividing the values into five categories and assigning each category with an integer number starting from 1 through 5. These integer values correspond to different levels of relative importance starting from very high (1) to very low (5). For example, we normalize building footprints such that streets with higher building footprints have lower weights. The reason is that a higher building footprint value indicates that the street is located in a high-density residential area or a business center, therefore the stormwater should be drained quicker. We note that the only requirement for a normalized weight is that it should be greater than zero since zero edge weights may lead to having infinity paths with equal lengths, thus the shortest path cannot be determined. After transforming all the attributes into edge weights by normalizing them to the range [1, 5], we compute the integrated weight of edges by taking the average of four weights. The same relative importance logic applies to the integrated weight, i.e., a lower integrated weight value for a street increases the probability of the street having a higher BC value. Consequently, the street has a higher relative significance and requires more stormwater transport capacity. We provide more details on the implementation of these rules in our algorithm, in the following section.

### 2.1.2 Generic procedure for deriving BUSN

In this subsection, we outline a generic procedure to derive BUSN based on IWBC that can be conceptually applicable to any urban area. Figure 6 illustrates the major steps in this generic procedure, which are explained below.

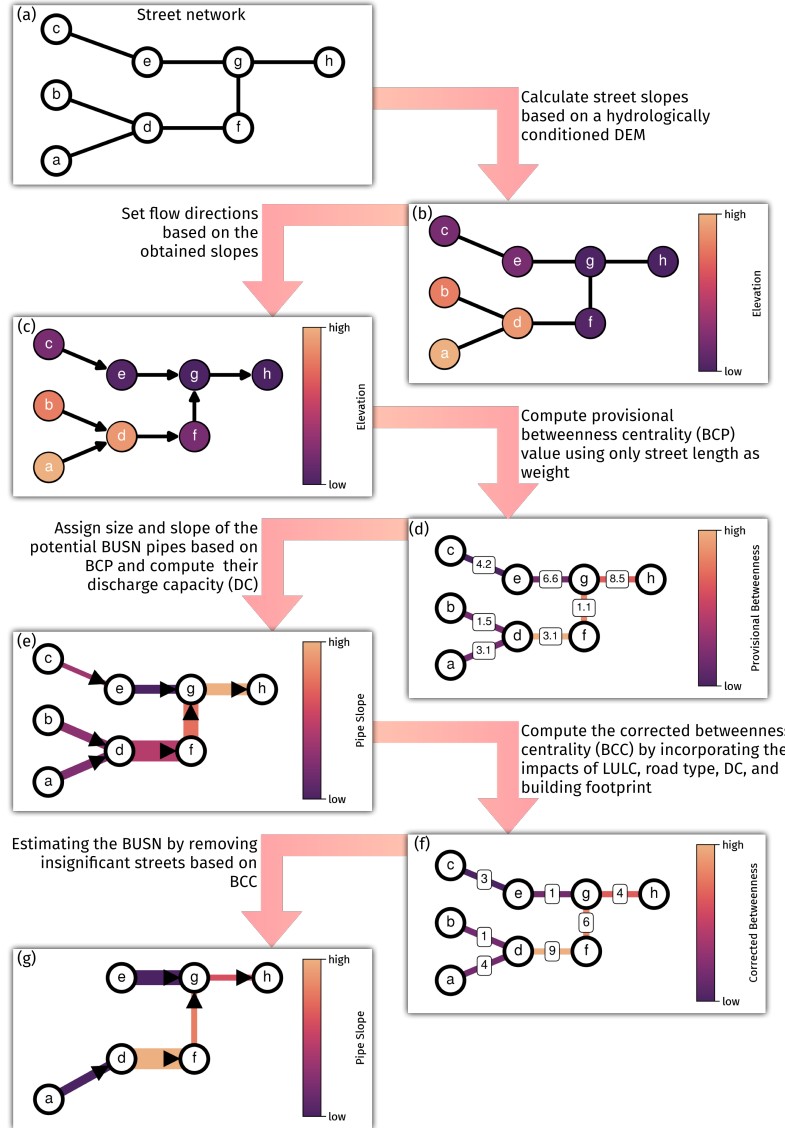

**Figure 3.** A schematic diagram of a generic procedure for estimating BUSNs using publicly available above-ground datasets.

1. Calculating surface slopes of all streets in the street network of interest (Figure 3a) based on the available Digital Elevation Model (DEM) (Figure 3b) data. The DEM data should ideally have the highest spatial resolution and a small error

in vertical elevation values when possible, to minimize the errors in estimating street slopes. When necessary, Lidar data can be used instead for better accuracy.

2. Setting flow directions between streets based on the surface slopes obtained in Step 1. At this stage, we assume street length is the only weighting factor. Now the street network becomes a directed, weighted network, as shown in Figure 3c.

3. Initial estimation of relative street importance by calculating the weighted betweenness, as shown in Figure 3d. In this study, we use two Python packages for performing these operations: Networkit (Staudt et al., 2015) for running computationally expensive network operations such as computing BC in parallel and Networkx (Hagberg et al., 2008) for other network operations such as community detection and measuring network connectivity.

4. Estimating streets' Right-Of-Way (ROW) based on local/federal regulations. ROW is a part of the land that is reserved by local/federal authorities for construction, maintenance, and future expansion of transportation elements such as highways and public utilities (Brown et al., 2013).

5. Estimating the hydraulic properties of the potential BUSN pipes, e.g., pipe size and slope, by accounting for both the weighted betweenness from Step 3 and the recommendations from the street/road relevant regulations at the federal, state, or local levels (see Figure 3e). Steps 4 and 5 are where the site-specific factors come into play since different cities under different jurisdictions may have site-specific requirements on ROW and the hydraulic properties of BUSNs.

6. Calculating IWBC by assigning different weights to different streets and integrating several weighting factors such as road type, land use/land cover (LULC), surface topography, and building density.

7. Deriving BUSN by removing those relatively unimportant streets based on IWBC. We assume BUSN pipes are only needed for those remaining streets. The topology of the BUSN is thus the same as that of the remaining, relatively important streets.

8. Checking the connectivity of the remaining network based on the concept of Weakly Connected Components from Graph theory. In Graph theory, network connectivity is an important measure of a network's resilience to losing edges or nodes, i.e., the impact that removing edges and nodes have on the overall network flow. For this purpose, after removing the unimportant streets, first, we detect the isolated subnetworks by determining the weakly connected components, i.e., those components that are unreachable after converting the network to an undirected graph by ignoring edge directions. Then, we find the number of streets for each subnetwork and remove those subnetworks whose number of streets is less than the average street count of the subnetworks.

In Step 5, we proposed a weight integration strategy for combining continuous and discrete weighting factors into a unified discrete weight system. Some urban features, such as road type and land use/land cover, are only quantified with discrete values and cannot be represented by continuous values. The integrated weight ranges from one to five. For any edge, a smaller weight indicates higher relative importance since the edge will have a higher chance of being on the shortest path. First, we transform

all continuous weighting factors into discrete values in two possible ways: 1) the quantile method, which is based on the equal number of features in each class, and 2) the Fisher-Jenks (Jenks, 1977) method, which minimizes the total within-class variance

of features. Upon transforming all weighting factors to discrete values, we integrate them via arithmetic mean. Note that, by using arithmetic mean, we equally account for different weighting factors since no data are available for objectively quantifying the relative importance of each weighting factor.

### 2.1.3 Validation

Due to the scarcity of publically-available real BUSN data and their low quality, we can only validate the topology of the derived

BUSNs. Therefore, although our proposed algorithm provides hydraulically feasible approximations for the size and slope of the BUSN pipes, we cannot validate them. Our topology validation strategy is based on the principle of spatial proximity for the places where some real BUSN data are available. As shown in Figure 4, on each side of a derived BUSN edge, we set a buffer zone as wide as the corresponding ROW. We then perform spatial analysis to judge how much of a pipe from real BUSN is located within the buffer zones. If 60% of a pipe length from the real BUSN is within this buffer zone, the pipe is considered

"covered". We determined this 60% threshold based on a sensitivity analysis, which shows that the total coverage percentage value does not change more than 2% for threshold values from 50% to 80%. For any area, we denote $L_{\text{covered}}$ as the total length of the "covered" BUSN pipes, and $L_{\text{all}}$ as the total length of all the BUSN pipes. To measure the algorithm's performance over an urban area, we define a successful covering percentage as

$$\omega = \frac{L_{\text{covered}}}{L_{\text{all}}} \cdot 100. \tag{2}$$

Obviously, the higher the $\omega$ value, the higher the percentage of the real BUSN pipes successfully covered by the algorithm, and thus the better the algorithm's performance is.

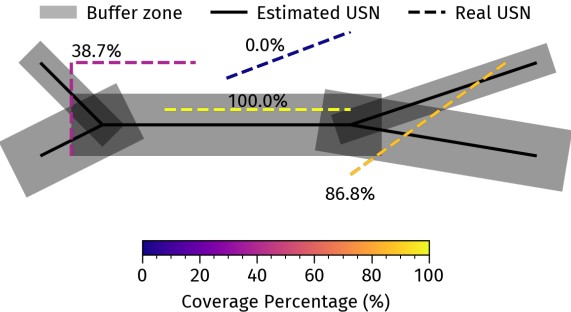

**Figure 4.** A simple case demonstrating the validation method. Real BUSN elements with more than 60% coverage percentage are considered as "covered".

There are often non-negligible uncertainties in both street network and real BUSN data. For instance, over any urban area, one may estimate both the total length values of the real BUSN pipes and the streets, respectively, from the available data.

In principle, the real BUSN's total length should not exceed that of the corresponding street network. In reality, however, this
may not be the case if there are much fewer missing data from real BUSN data than the corresponding street network data. To
account for this situation in our validation, first, we discretize the targeted urban domain into 1 km resolution grid cells. Then,
for each grid cell, we calculate the ratio of the total length of the real BUSN pipes to the total length of the street network
elements. Theoretically, this ratio should be no more than 1.0. We discard those cells with this ratio larger than the theoretical
maximum, 1.0, and only calculate $\omega$ in the remaining grid cells. Finally, the spatial average of $\omega$ values over the grid cells
within an urban area ($\overline{\omega}$) is the metric we use to measure the algorithm's performance for this area.

## 2.2 Technical implementation of the U.S.

This subsection describes the detailed implementation of the algorithm outlined previously, over the U.S.

### 2.2.1 Data Gathering and Processing

We retrieve the raw input data that are available at least in the U.S. from the following sources:

- Street network data from OpenStreetMap (OpenStreetMap, 2021) using an open-source Python package called OSMnx
  (Boeing, 2017). OSMnx retrieves street network data and their attributes such as road type and street length for any
  region of interest and can perform some post-processing operations for cleaning up the raw street network data.

- Digital Elevation Model (DEM) at a 10 m resolution from the 3D Elevation Program (3DEP) of the U.S. Geologic Survey
  (USGS) (U.S. Geological Survey, 2017) using an open-source Python package called Py3DEP (Chegini et al., 2021).
  Py3DEP provides access to the highest quality and resolution DEM data that are publicly available in the US.

- Land use/land cover (LULC) data at 30 m resolution from the Multi-Resolution Land Characteristics (MRLC) consor-
  tium (Dewitz and U.S. Geological Survey, 2021) using an open-source Python package called PyGeoHydro (Chegini
  et al., 2021). This package provides access to many hydroclimate datasets such as streamflow observations from USGS
  stream gauges and the National Inventory of Dam, in addition to LULC data.

- Building footprints from the Microsoft Building Footprints (MSBF) (Microsoft, 2018) dataset using an open-source
  Python package called PyGeoOGC (Chegini et al., 2021). Building footprints are the perimeter of buildings that out-
  lines their exterior walls. In total, this dataset includes about 130 million building footprints over the U.S. Moreover,
  PyGeoOGC is a low-level interface to many geospatial web services.

- BUSN design criteria and recommended parameters from the Urban Drainage Design Manual by the Federal Highway
  Administration, U.S. Department of Transportation (Brown et al., 2013), and other state or local governments (.e.g.,
  UDFC, 2018).

We perform the following post-processing operations on the raw input data:

- We retrieve the "Road type" and "length" attributes for each street directly from OpenStreetMap (see Table 2).

- We calculate the surface slope and flow direction of each street in four steps: 1) Within any street network, removing intersection points that are closer than the DEM resolution, therefore, cannot be effectively used; 2) Hydrological conditioning of DEM data to more accurately represent flow direction of surface runoff; 3) Computing street slope using the conditioned DEM data, and setting the slope as 0.4% if the computed slope value is less than 0.4% since any streets must have a minimum longitudinal slope of 0.4% (UDFC, 2018); 4) Setting flow directions between streets (we assume the flow directions in the underlying BUSN pipes are the same as the streets).

- We estimate the streets' ROW in four steps: 1) Assigning the number of lanes to each street based on its road type defined in OpenStreetMap (OpenStreetMap, 2021) (see Table 2). Since the remaining road types that are not listed in Table 2 are mostly local access roads we assign them two lanes; 2) Setting the width of an individual lane as 5.0 m, which is based on the minimum recommended street lane width of 3.6 m and the minimum sidewalk width of 1.5 m (UDFC, 2018); 3) Calculating the total width of each street as the multiplication of the number of lanes and lane width; 4) Assigning one buffer zone on each side of the street and as wide as the street itself.

- We determine the dominant land cover type in the buffer zone of each street by computing the dominant cover type within the buffer zone from the high-resolution LULC data.

- We estimate the total area of building footprints within the buffer zone of each street by summing up the footprints of the buildings with more than 30% of their areas within the buffer zone.

Upon performing these post-processing operations, each street has seven attributes: road type, length, ROW, surface slope, flow direction, land cover type, and building footprints' area. Once all the input data are ready, we consider four weighting factors for IWBC: road type, LULC, building footprint area, and stormwater pipe flow capacity, as shown in Table 3. We note that in this study we transform these street attributes into edge weights in such a way that higher IWBC values correspond to higher significance for transporting stormwater. Here stormwater pipe flow capacity ($Q_f$) is the maximum discharge for a pipe full of water (more details are provided later). These four weighting factors are chosen for two reasons: 1) they all impact urban stormwater transport; 2) they can be estimated in the U.S. from the existing data. Other weighting factors may also impact urban stormwater, but there is insufficient supporting data to translate them into quantitative weight.

We group each weighting factor into five classes and assign a provisional weight to each class. We set a higher provisional weight to a class with less importance and vice versa. Recall that, for any edge in a weighted network, a higher weight implies a lower probability to be included in the shortest path and thus a smaller IWBC value.

The weighting factors in Table 3 include both discrete (road type and LULC) and continuous (building footprints and pipe flow capacity) variables. For consistent weight calculation, we group each weighting factor into five classes. Road type and LULC are already discrete, so the grouping is straightforward. For building footprints and pipe flow capacity, we use the Fisher-Jenks natural breaks method to group them into five classes. We then assign provisional weights to these classes, i.e., a higher provisional weight to a class with less importance and vice versa. Recall that, for any edge in a weighted network, a higher weight implies a lower probability to be included in the shortest path and thus a smaller IWBC value. This way, each street will have four provisional weights, and its final weight is taken as the arithmetic mean of these provisional weights.

**Table 2.** Road types' definitions (OpenStreetMap, 2021) and their corresponding number of lanes.

| Road Type | Definition* | No. of Lanes |
|---|---|---|
| Motorway | A restricted access major divided highway, normally with 2 or more running lanes plus emergency hard shoulder. Equivalent to the Freeway, Autobahn, etc. | 6 |
| Trunk | The most important roads in a country's system that aren't motorways. (Need not necessarily be a divided highway.) | 4 |
| Primary | The next most important roads in a country's system. (Often link larger towns.) | 4 |
| Secondary | The next most important roads in a country's system. (Often link towns.) | 3 |
| Tertiary | The next most important roads in a country's system. (Often link smaller towns and villages) | 2 |
| Residential | Roads which serve as an access to housing, without function of connecting settlements. Often lined with housing. | 2 |
| Busway | A dedicated roadway for bus rapid transit systems. | 1 |
| Link | The link roads (sliproads/ramps) leading to/from a higher class road from/to the same or lower class toad. | 1 |

*  Definitions are direct quotes from the OpenStreetMap wiki (© OpenStreetMap contributors, 2022)

**Table 3.** Ranges and scores of the weighting factors

| Score | Land Cover Type | Road Type | Building Area* ($m^2$) | Pipe Flow Capacity* ($m^3/s$) |
|---|---|---|---|---|
| Very high (1) | 24 | Residential and tertiary | $>J_4$ | $>J_4$ |
| High (2) | 23 | Secondary | $J_4–J_3$ | $J_4–J_3$ |
| Moderate (3) | 22 | Primary | $J_3–J_2$ | $J_3–J_2$ |
| Low (4) | 21 | Motorway and trunks | $J_2–J_1$ | $J_2–J_1$ |
| Very low (5) | Other types | Other types | $<J_1$ | $<J_1$ |

$J_i$ corresponds to bin values obtained from Fisher-Jenks natural breaks' classification algorithm

We derive $Q_f$ from the existing information using Equation 3 (Thomason, 2019). This equation is for a circular concrete pipe with a diameter of less than 24 in. (0.6 m) which is one of the most commonly used stormwater pipes in the U.S. (Heilman, 2019).

$$Q_f = \frac{0.3116}{n} D^{8/3} S^{1/2}, \tag{3}$$

where $n$, $D$, and $S$ are the pipe's Manning's roughness coefficient, diameter (m), and slope (m/m), respectively. Although the maximum discharge capacity of a circular pipe occurs at 94% of a pipe diameter (Chow, 1959), it does not make a difference in our proposed algorithm since we are only concerned with the relative discharge capacity of pipes in the network. The recommended $n$ value for stormwater pipes with a diameter less than or equal to 24 in. (0.6 m) is 0.013 (Heilman, 2019). $D$ and $S$ are determined during a two-stage, predictor-corrector procedure to compute the IWBC values as follows:

- Compute provisional BC values ($\hat{BC}$) for the entire network using Equation 3 by only considering length as the street weight.

- Assign a suitable pipe size to each street in the network based on the $\hat{BC}$ values. We achieve this by statistical binning of these values into 10 intervals (the number of permissible pipe sizes from Table 4). Considering that streets with higher $\hat{BC}$ values are likely to receive more stormwater, the largest pipe size is assigned to streets in the class with the highest $\hat{BC}$ values. Similarly, the smallest pipe size is assigned to streets in the class with the lowest $\hat{BC}$ values.

- Set pipe slopes based on the obtained street slopes and the permissible slope ranges corresponding to each pipe size given in Table 4 adopted from Brown et al. (2013). A pipe slope usually follows the street slope unless the slope is outside the permissible range since it can lead to pipe flow velocities below or above the design criterion (0.9–3 ms$^{-1}$). Therefore, if a street slope is less than the minimum permissible pipe slope, we set the pipe slope to its minimum permissible value. Similarly, if a street slope exceeds the maximum permissible pipe slope, we set the pipe slope to its maximum permissible value. Subsequently, we compute the pipe flow capacity using Equation 3 and the obtained pipe sizes and slopes.

- Compute the arithmetic mean of the four weighting factors and determine the corrected BC values for the network, i.e., the final IWBC values.

Note that this predictor-corrector approach does not yet change the baseline street network topology. The obtained IWBC values are our basis to obtain the derived BUSN by removing those relatively unimportant streets from the baseline network. One may begin by removing those edges with their IWBC values less than a threshold since these edges represent those less important streets that are less likely to require below-ground stormwater pipes. Intuitively, by increasing the IWBC threshold, we remove more elements from the baseline street network, thus the drainage capacity of the BUSN corresponding to the remaining part of the street network decreases. There is nevertheless a nonlinear relationship between increasing the IWBC threshold and decreasing derived BUSN. This is because of two reasons:

**Table 4.** Slope range based on storm drain pipe size

| Pipe Size (in.) | Minimum Slope (%) | Maximum Slope (%) |
|:---:|:---:|:---:|
| 12 | 0.220 | 4.860 |
| 14 | 0.170 | 4.000 |
| 15 | 0.150 | 3.610 |
| 16 | 0.140 | 3.310 |
| 18 | 0.120 | 2.830 |
| 21 | 0.100 | 2.300 |
| 24 | 0.080 | 1.930 |
| 27 | 0.067 | 1.650 |
| 30 | 0.058 | 1.430 |
| 36 | 0.046 | 1.120 |

– In most street networks, the numbers of edges associated with lower IWBC values are nonlinearly larger than those with higher IWBC values. Our analysis shows that the lowest IWBC values have the highest frequency in the network. For example, Figure 5 compares the distribution of IWBC values using two classification methods, namely Fisher-Jenks and Quantile, for the four cities that are subject of this study. As is evident from the figure, the first class based on the Fisher-Jenks method has a significantly higher edge count whereas the last class based on the Quantile method has significantly higher within-class variance in IWBC values.

– The edges with lower IWBC values are corresponding to the pipes with smaller diameters, and their removal has a smaller impact on the total BUSN's drainage capacity than removing those edges corresponding to the pipes with larger diameters.

Therefore, IWBC cannot be directly used to guide this removal operation. We carry out this operation in two steps:

– We use the first out of ten classes based on the Fisher-Jenks method (FJ1) to identify the group of streets with the lowest IWBC values. FJ1 has the highest edge count and the lowest within-class variance in IWBC values.

– Considering the small variance of IWBC values of the streets in FJ1, the Quantile method is suitable for categorizing the streets based on their IWBC values. Thus, we use the Quantile method as an indicator for removing edges from the baseline network.

Our empirical analysis of the real BUSN data from the U.S. cities suggests that, for a baseline street network, those edges with their IWBC values belonging to the FJ2 to FJ10 classes are important enough to have underlying BUSN pipes, whilst only a fraction of the edges with their IWBC values belonging to the FJ1 class do not have underlying BUSN pipes and should be removed. We hence define the Drainage Adequacy Classifier Index (DACI) based on the IWBC quantiles within the FJ1 class.

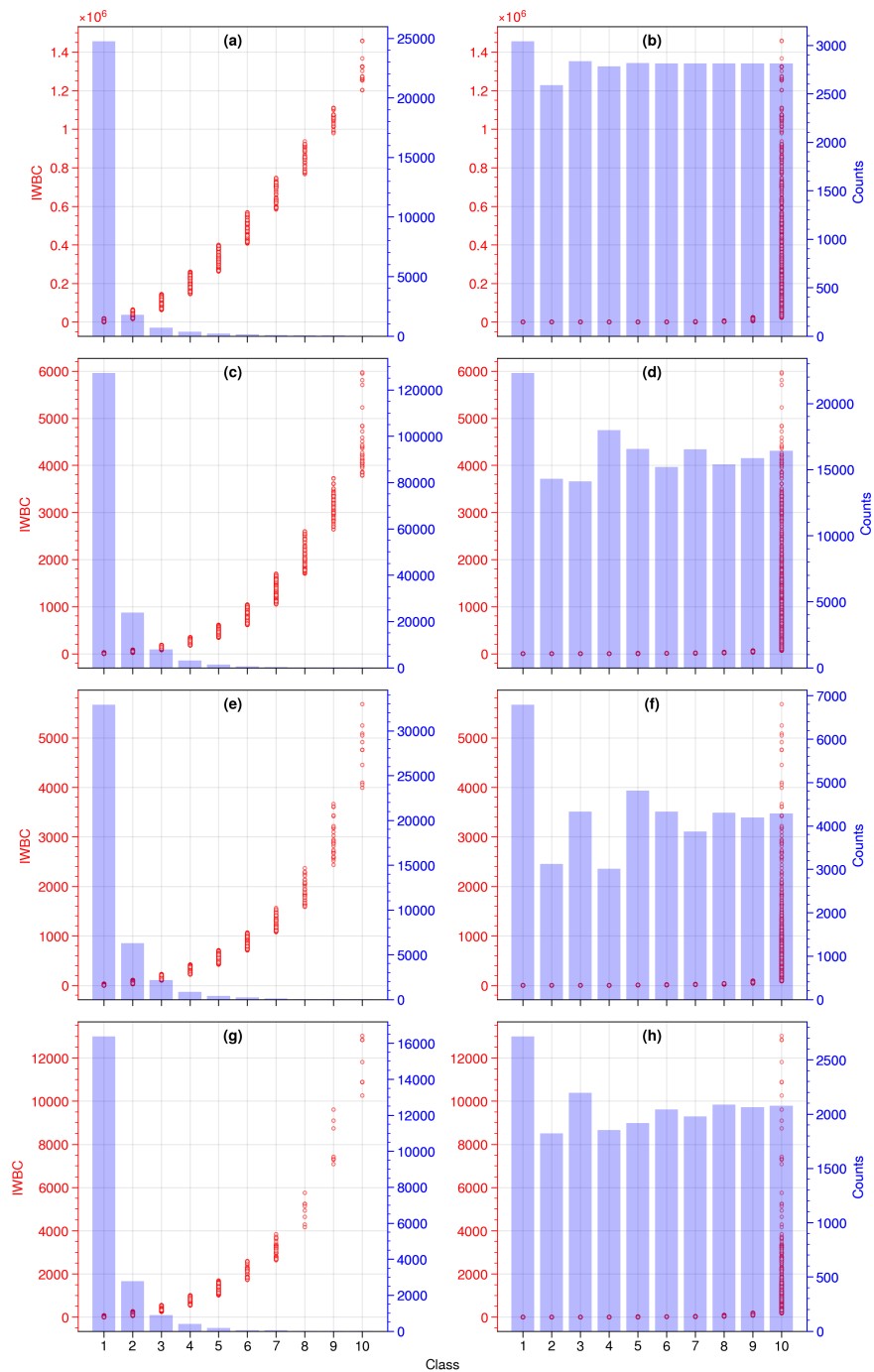

**Figure 5.** Comparing Fisher-Jenks (plots on the left column) and Quantile (plots on the right column) classification methods for IWBC values of baseline networks. (a, b), (c, d), (e, f), (g, h) correspond to the Los Angeles, Houston, Baltimore, and Seattle cases, respectively. In each plot, IWBC values are on the left $y$-axis (red) and the number of elements in each class is on the right $y$-axis (purple).

Mathematically, DACI = 1 - IWBC quantile. We use DACI as a direct indicator in guiding the removal of relatively unimportant edges from a baseline street network and deriving the final BUSN. For example, if the DACI value is 0.8, we drop those edges with their IWBC values less than the 20th-quantile of the IWBC values in the FJ1 class. A DACI value of 1.0 suggests retaining all the edges in a baseline network.

In this study, we use DACI as an empirical parameter. For a case study where real BUSN data are partially available, we increase the DACI value until there is no significant increase in the average $\omega$ value. Figure 6 depicts a flowchart of our proposed framework.

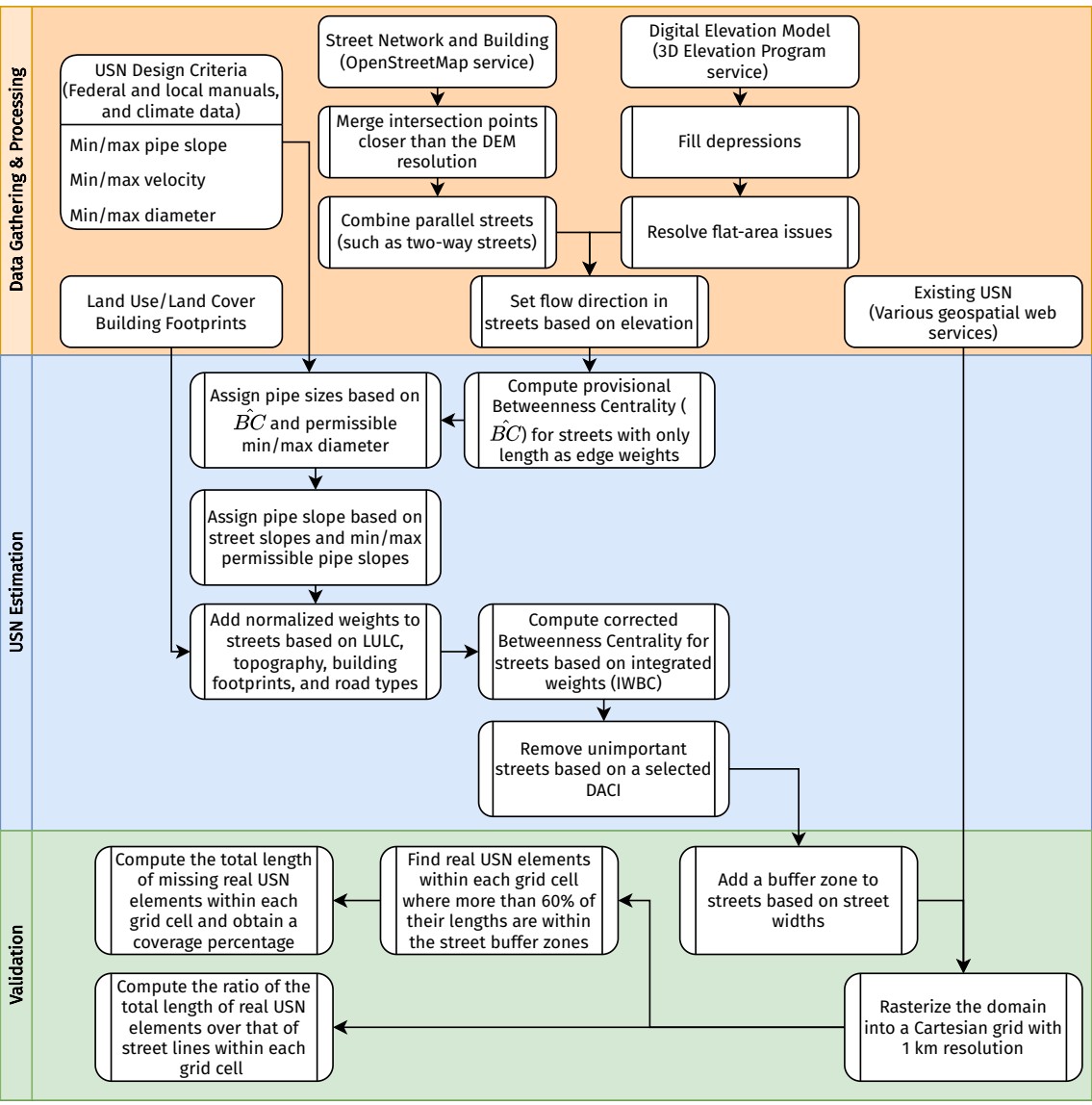

**Figure 6.** A flowchart demonstrating details of the proposed framework

## 3   Results

### 3.1   Case Studies and data

We choose four major cities in the U.S., including Houston, TX, Los Angeles, CA, Baltimore, M.D., and Seattle, WA, as the case studies to demonstrate the algorithm (Figure 7). The primary reason for choosing them is that there are real BUSN data available to the public over a fraction of areas within these cities with relatively decent data quality. For instance, over the Los Angeles metropolitan area, we only obtain the real BUSN data for a small town, San Fernando Valley.

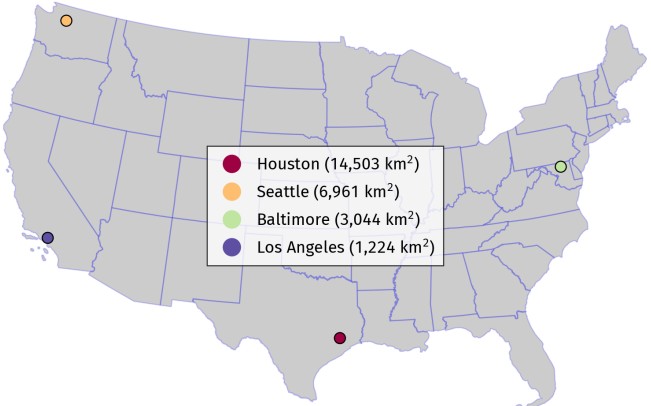

**Figure 7.** A map of urban areas that are subject of this study with their area.

Interestingly, the real BUSN and street network data show different characteristics among these case studies. The street network structure in the Baltimore and Seattle cases is quite different from that in the Houston and Los Angeles cases. In Graph theory, a community refers to a group of nodes (street intersections) in a network where the density of the connections among them is higher than the rest of the network. In a street network, a community can be analogous to an urban cluster. Table 5 summarizes some real BUSN and street network data for our four case studies. As is evident from the real BUSN columns of the table, the quality, availability, and types of real BUSN data vary case by case since local authorities produce them based on their own rules, regulations, and available resources. The table shows that the number of element type categories in each case is different. For example, in the Houston case, BUSN element types are divided into seven categories (Trunk, Lead, Outfall, Culvert, Trench Drain, Siphon, Overflow) whereas in the Seattle cases there are four categories (Detention Pipe, Driveway Culvert, Cross Culvert, Pipe). Thus, not only the data quality but also the level of details that each real BUSN dataset provides is different. Nevertheless, as is evident from Table 5, the main storm drain pipes (sometimes called trunks), which are the focus of this study, are the most dominant elements in BUSNs, both in terms of quantity and their role in the system's transport capacity. Moreover, according to Table 5 the Baltimore and Seattle cases have smaller average community sizes and thus smaller urban clusters.

**Table 5.** Summary of real BUSN and street network data

| Case | Real BUSN | | | Street | | |
|------|-----------|---|---|--------|---|---|
| | Total Length (km) | # of Types | Trunk (%) | Total Length (km) | # of Comm. | Ave. Comm. Size |
| Los Angeles | 1,612 | 12 | 79.2 | 5,668 | 97 | 205.0 |
| Houston | 6,352 | 7 | 71.7 | 41,989 | 266 | 538.1 |
| Baltimore | 2,461 | 10 | 94.7 | 9,946 | 126 | 258.2 |
| Seattle | 1,857 | 4 | 83.5 | 5,189 | 128 | 144.7 |

"# of Types" means the number of element type categories in a database, e.g., trunk and culvert. and "Comm." stands for network communities.

We retrieve the input data for the four case studies following the procedure described in Section 2.2.1. Figure 8 shows an example of the input data that we collected for the Los Angeles case. Figures 8a and 8b plot the Digital Elevation Model and land use/land cover data that we retrieved from 3DEP (U.S. Geological Survey, 2017) and MRLC (Dewitz and U.S. Geological Survey, 2021), respectively. Additionally, Figures 8c, 8d, and 8e represent the street network, existing BUSN, and building footprints that we obtained from OSM (OpenStreetMap, 2021), Los Angeles GeoHub (Los Angeles GeoHub, 2022), and MSBF (Microsoft, 2018), respectively.

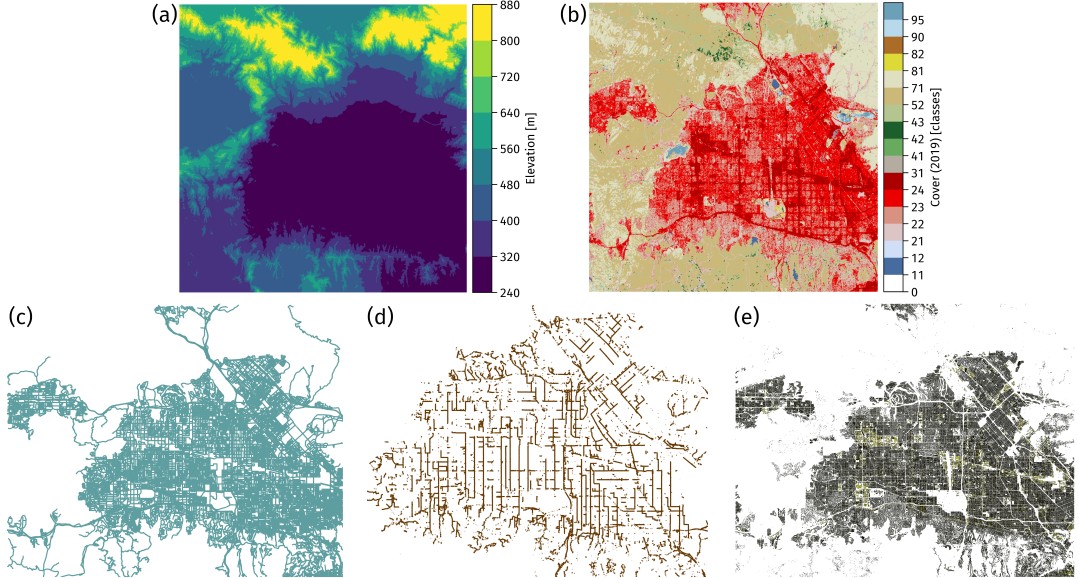

**Figure 8.** Input data for the Los Angeles case: (a) Digital Elevation Model, (b) land use/land cover, (c) street network, (d) existing BUSN, and (e) building footprints

Furthermore, although the input data for the proposed BUSN algorithm are generally available in the U.S., the data quality might vary among different categories of input data and different locations. For example, in the available real BUSN data, there

are often some missing edges. As is evident from Figure 9a, the BUSN data is only available in some urban areas in Houston, as indicated by the red color. However, in the other areas, there should be BUSNs except that the real data are not available. Our

observation suggests that different case studies may be subject to different levels of real BUSN data quality issues, which lead to some uncertainty in the validation of the derived BUSN. Moreover, it appears that the quality of street network data is not consistent across locations either. Figure 9b shows the street network data from OpenStreetMap overlaid with that from Google Maps for a tiny portion of Seattle, WA (for better clarity), suggesting that OpenStreetMap misses a considerable number of streets and thus has certain data quality issues as well. Since the topology of our derived BUSNs is primarily based on that of

the underlying street networks, the accuracy of the derived BUSN data is also subject to the quality of the underlying street network data.

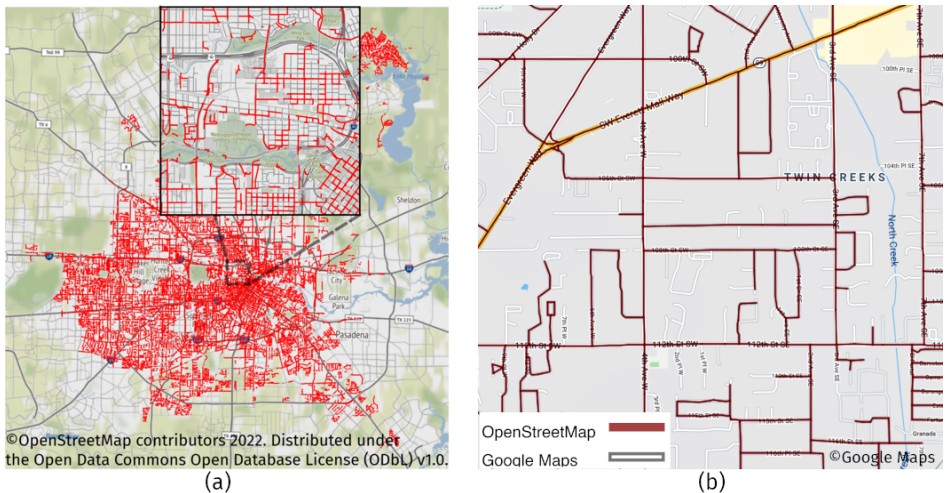

**Figure 9.** Examples demonstrating the quality of publicly available data sets. (a) Existing BUSN data for Houston is published by Houston Public Works (City of Houston, 2021), and (b) street data for a part of Snohomish, Seattle, WA, from OpenStreetMap (OpenStreetMap, 2021) and Google Maps (commercial) (Gorelick et al., 2017), (© OpenStreetMap contributors YEAR. Distributed under the Open Data Commons Open Database License (ODbL) v1.0. and © Google Maps)

## 3.2 BUSN Derivation and Validation

For each of the case studies, we run the algorithm with DACI values varying from 0.0 and 1.0 with a 0.05 interval (Figure 10). Intuitively, one would expect that $\overline{\omega}$ increases with DACI. Interestingly, there is a plateau behavior for the Baltimore, Seattle,

and Houston cases, where $\overline{\omega}$ stops increasing once DACI passes a threshold value, i.e., $\overline{\omega}$ reaches its maximum and stabilizes. Recall that a higher DACI value implies that more streets in a street network require underlying stormwater pipes. This plateau behavior confirms our earlier statement that those relatively unimportant streets will not require underlying stormwater pipes. The DACI thresholds (for this plateau behavior) are 0.9, 0.9, and 0.8 for the Baltimore, Seattle, and Houston cases, respectively, suggesting that in Houston there are more unimportant streets that do not require underlying stormwater pipes than the

Baltimore and Seattle cases. The Los Angeles case, however, does not seem to have such a plateau behavior, i.e., $\overline{\omega}$ does not stop increasing with DACI even when DACI $= 1.0$. The possible reason is that there are stormwater pipes under most streets in the Los Angeles case. Therefore, as we retain more edges from the baseline street network (as the derived BUSN pipes), $\overline{\omega}$ keeps increasing. We note that since the street network data quality is sufficiently good for the Los Angeles case, this is not a reason for not reaching a plateau. We further confirm the street network data quality by retrieving higher-quality data from a
local source, Los Angeles GeoHub (2022), and comparing it with the data obtained from the OSM.

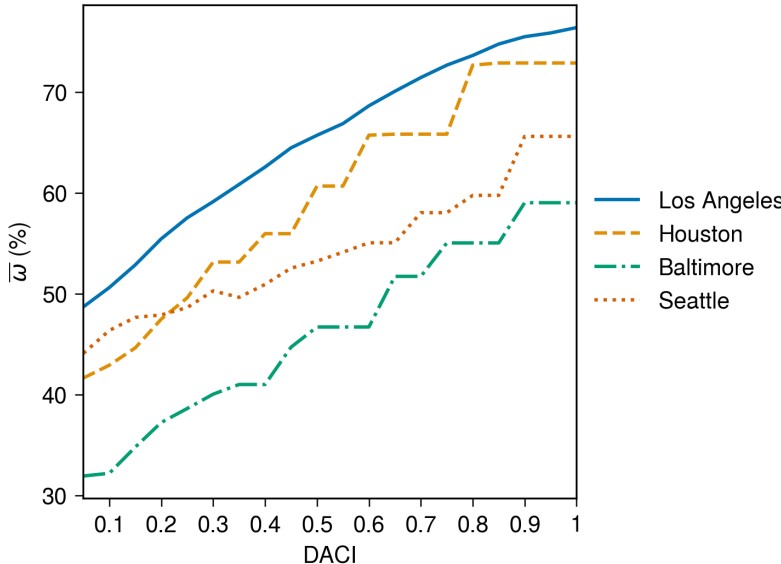

**Figure 10.** Model performance with DACI values varying from 0.0 and 1.0 with a 0.05 interval for the four case studies.

Figure 11 illustrates the changing spatial patterns of the baseline street network (Figure 11a) and the derived BUSNs when using the DACI value as 0.9, 0.7 and 0.5, respectively (Figure 11b-d), for the Los Angeles case. As expected, by decreasing the DACI value the derived BUSN becomes sparser while there are no small isolated subnetworks. We recall that in the eighth step of our general procedure, the algorithm removes the small isolated subnetworks if any. This step is particularly important
since the connectivity of a network dictates its overall transport capacity.

Figures 12-15 show results of the four cases. Each of these figures includes three panels. The first panel (Figures 12a-15a) show the $\omega$ values at the grid cell level. The 2nd panel (Figures 12b-15b) shows the ratio of the total length of the real BUSN pipes to the total length of the street network elements within each grid cell. Note that in the 1st panel, those grid cells with such a ratio exceeding 1.0 are highlighted with a red color and excluded from the validation due to the poor quality of street network
data. A spatial average of the $\omega$ values from the remaining grid cells is thus used as the algorithm's overall performance for each case study (see Table 6). The 3rd panel (Figures 12c-15c) shows the derived BUSN overlaid by the corresponding real BUSN.

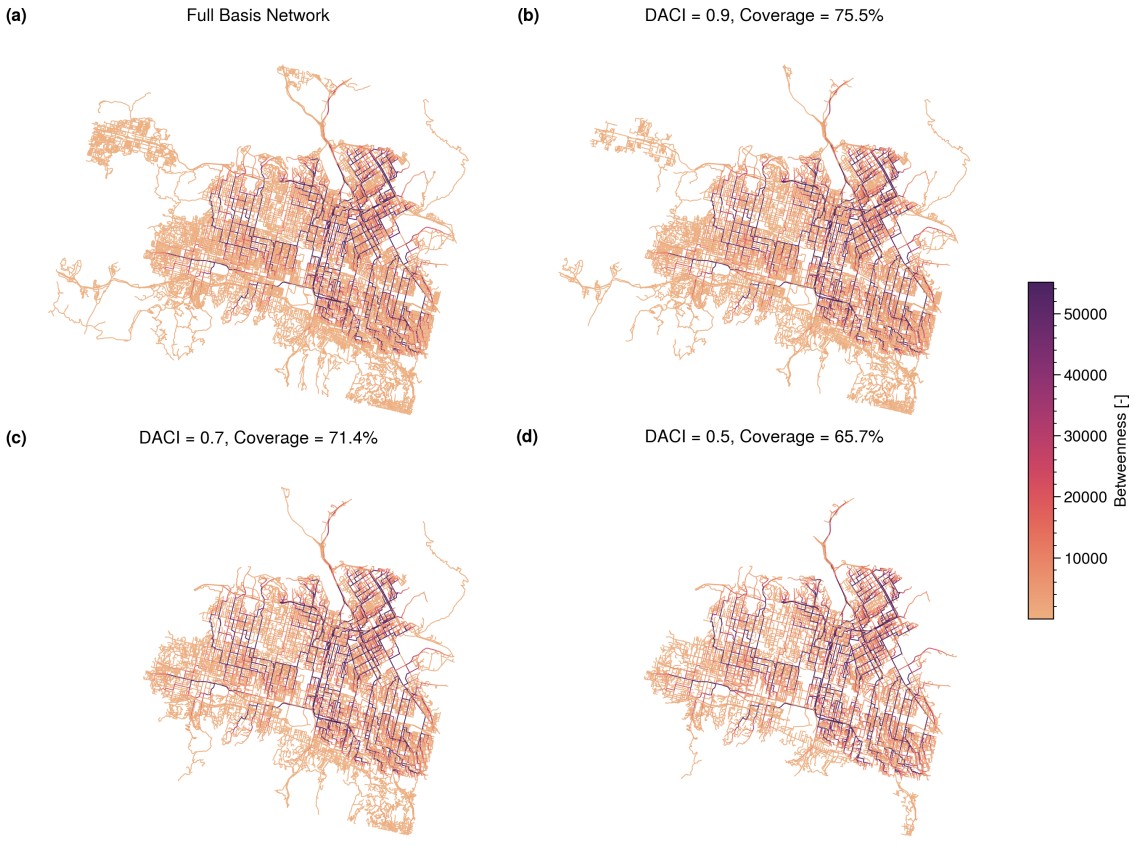

**Figure 11.** Three different BUSNs estimated for San Fernando Valley with different three different drainage discharge capacities; (a) Full street network, (b) 0.9, (c) 0.7, and (c) 0.5.

The algorithm performs very well in the Los Angeles and Houston cases, with $\overline{\omega}$ values 75.5% and 72.9%, respectively. It only performs reasonably well in the Seattle and Baltimore cases, with $\overline{\omega}$ values 65.6% and 59.0%, respectively. There are two possible reasons for this difference in the algorithm's performance: 1) The street network and real BUSN data quality is better in the Los Angeles and Houston cases than in the other two cases. In Figures 12b-13b, there are much less grid cells marked with the red color than Figures 14b-15b. 2) Topographic features can be attributed as another factor that affects the model performance since they can have a significant impact on the design and construction cost of below-ground drainage elements. Notably, the slope of streets can pose significant limitations on the construction of BUSNs. We recall that urban design manuals provide permissible slope ranges for BUSNs to maintain the pipes' flow velocity within a certain range. Therefore, urban areas in hilly terrains that can have higher street slope variability may require placing the BUSN pipes deeper into the ground and nonuniform cover depth (distance between the top of a BUSN pipe and the street surface). These requirements can lead to significantly higher construction costs since more excavation needs to be carried out.

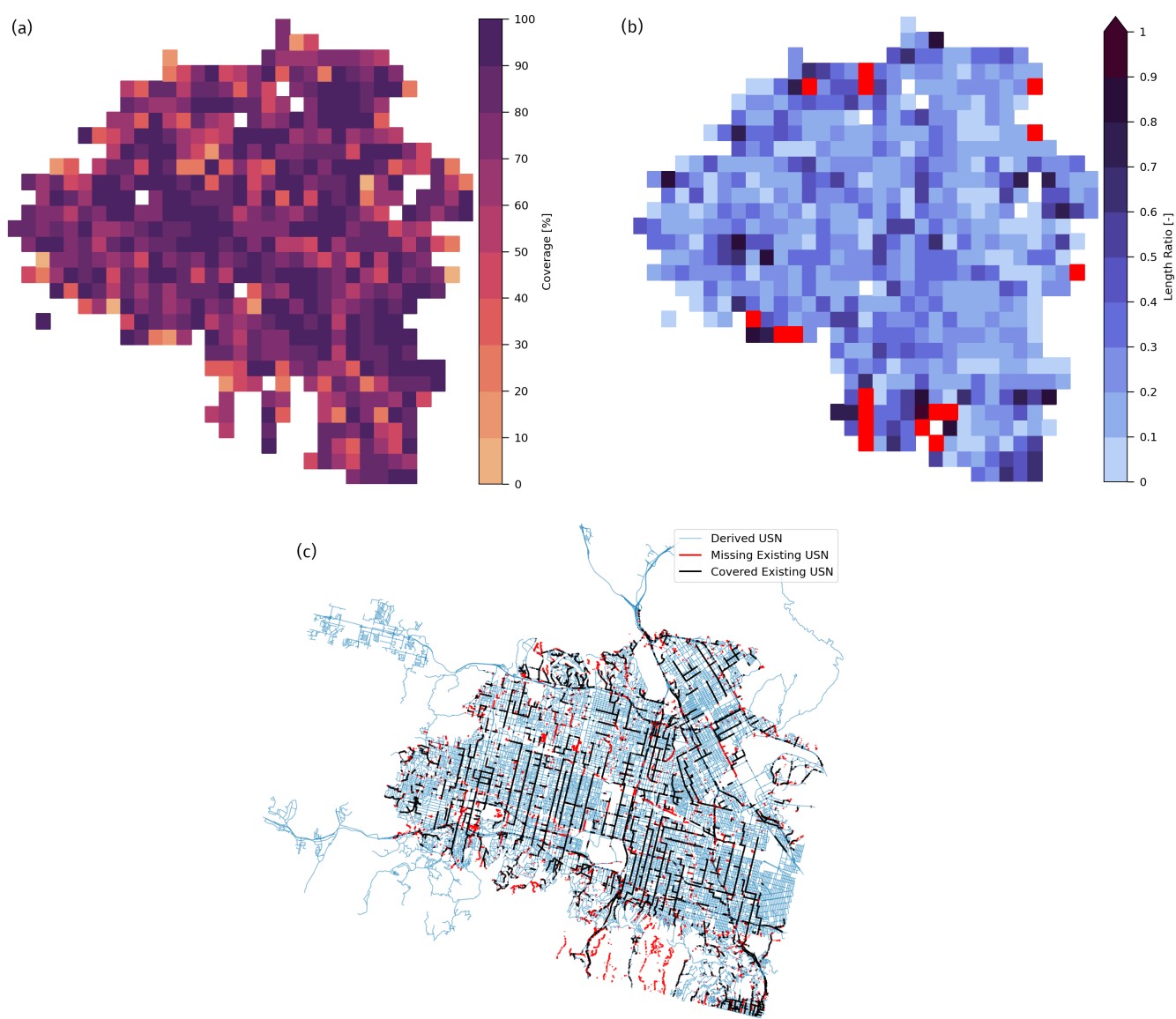

**Figure 12.** Results for the Los Angeles case. (a) The coverage percentage and (b) length ratio maps at 1 km resolution where the red pixels indicate the areas with poor street data quality. The white background pixels are grid cells that do not contain street and/or existing BUSN elements. (c) compares the estimated BUSN with the existing BUSN using 0.9 DACI.

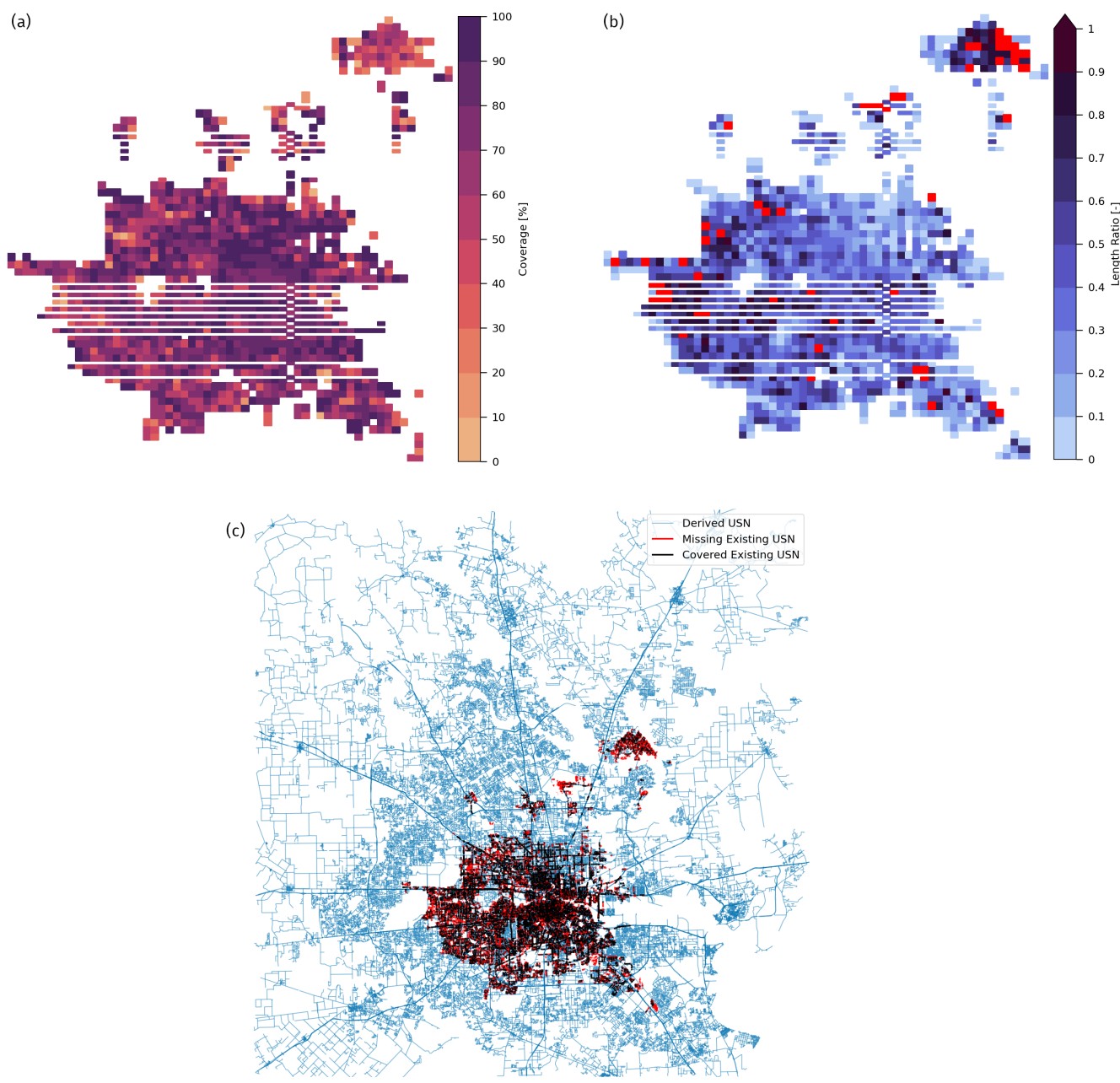

**Figure 13.** Results for the Houston case. (a) The coverage percentage and (b) length ratio maps at 1 km resolution where the red pixels indicate the areas with poor street data quality. The white background pixels are grid cells that do not contain street and/or existing BUSN elements. (c) compares the estimated BUSN with the existing BUSN using 0.8 DACI.

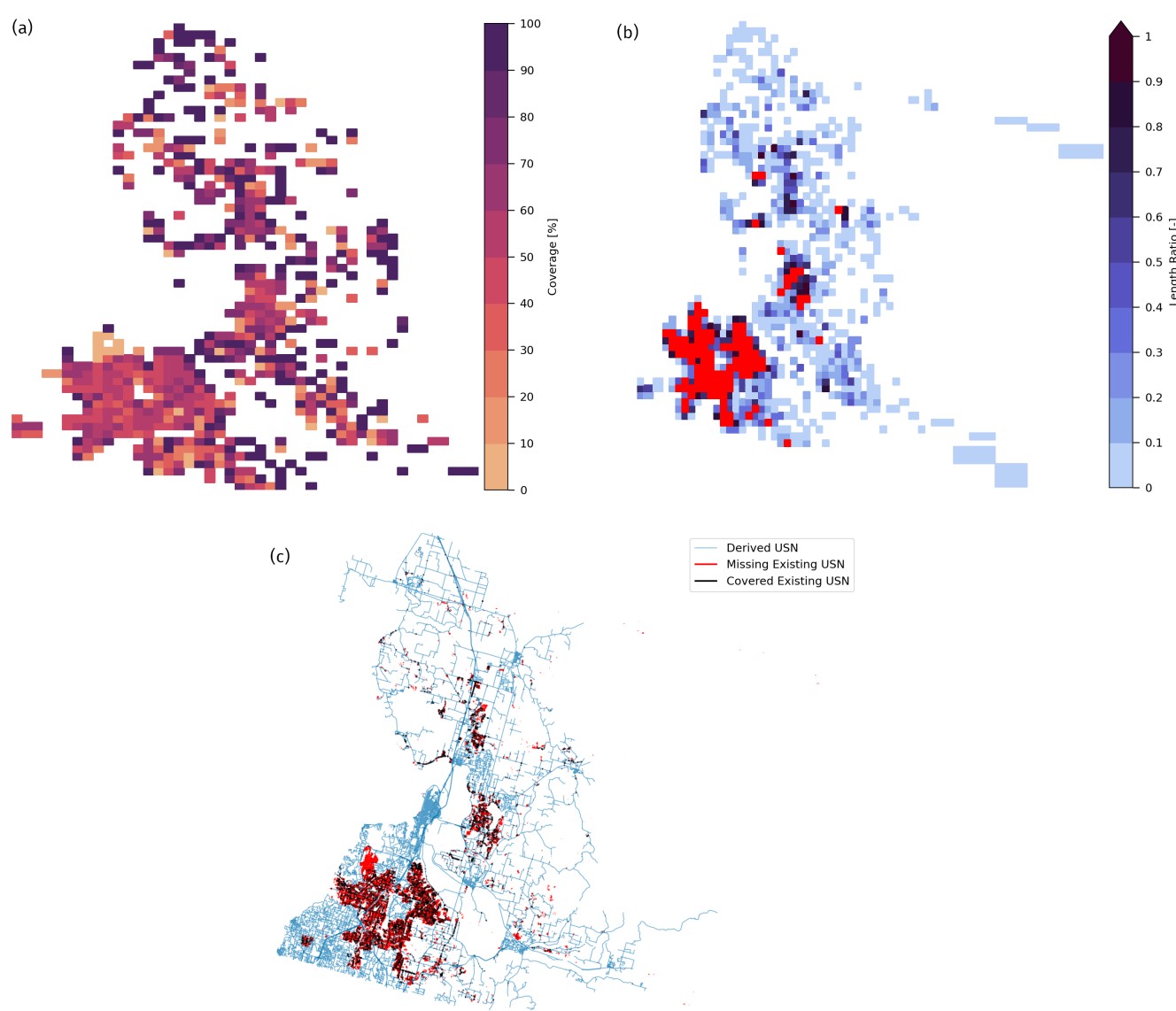

**Figure 14.** Results for the Seattle case. (a) The coverage percentage and (b) length ratio maps at 1 km resolution where the red pixels indicate the areas with poor street data quality. The white background pixels are grid cells that do not contain street and/or existing BUSN elements. (c) compares the estimated BUSN with the existing BUSN using 0.9 DACI.

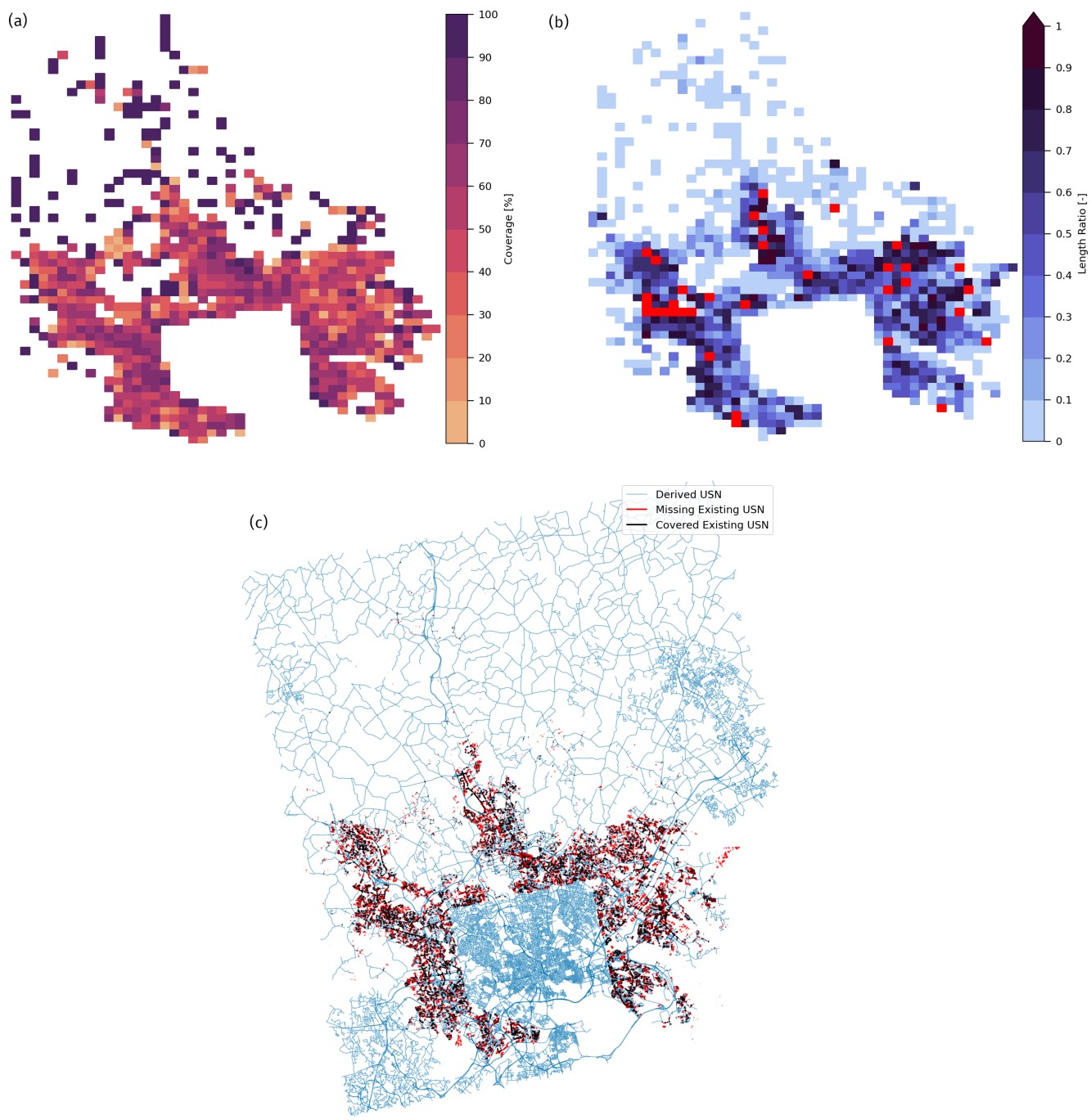

**Figure 15.** Results for the Baltimore case. (a) The coverage percentage and (b) length ratio maps at 1 km resolution where the red pixels indicate the areas with poor street data quality. The white background pixels are grid cells that do not contain street and/or existing BUSN elements. (c) compares the estimated BUSN with the existing BUSN using 0.9 DACI.

In this study, we are not accounting for the construction limitations and difficulties that arise in BUSNs in hilly terrains, therefore, we expect poorer performance in such areas. We quantify the slope variability of urban areas based on the Cumulative Percent (CP) graph for slope as shown in Figures 16a. For this purpose, as demonstrated in Figures 16b, we divide the slopes in the CP graph into four categories, namely, steep slopes with 0-5% exceedance probability, moderate slopes with 5-33% exceedance probability, mild slopes with 33-66% exceedance probability, and very mild slopes with 95-100% exceedance probability. Subsequently, we determine the slope variability by computing the gradient of the average slope values in each category, i.e., average slopes of the dashed lines in Figures 16b, as follows:

$$\overline{\gamma}_{ji} = \frac{1}{\Delta E_{ji}} \cdot \log\left(\frac{S_j}{S_i}\right) \cdot 100, \tag{4}$$

where $\overline{\gamma}_{ji}$ is the gradient from category $j$ to $i$ and $\Delta E_{ji}$ is the difference between exceedance probability percentages of consecutive categories, for example, $\Delta E_{33,5} = 33 - 5 = 28$. Moreover, $S_i$ is the average of all the slopes within category $i$. The reason that we use logarithm for computing the difference between consecutive average slopes is that the y-axis of the cumulative percent graph (Figure 16a) is log-scale. Higher gradients correspond to less slope variability.

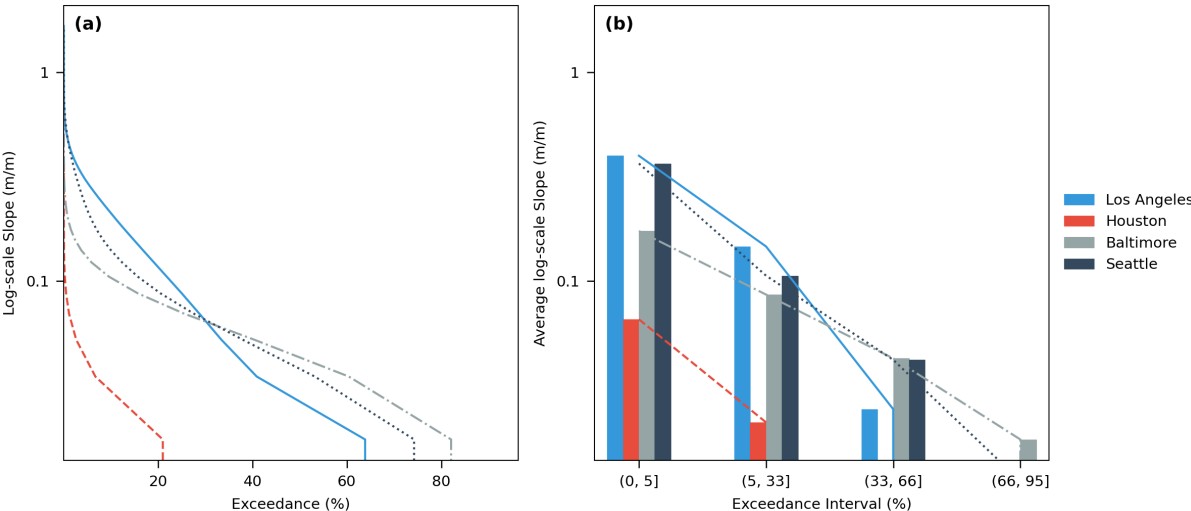

**Figure 16.** Comparison of the overland slope of urbanized areas for the four case studies based on (a) the cumulative percentage graph of slopes and (b) variability percentage of the slopes from steep slopes with 0-5% exceedance probability to moderate slopes with 5-33% exceedance probability, from moderate to mild slopes with 33-66% exceedance probability, and from mild to very mild slopes with 95-100% exceedance probability. The dashed lines in (b) represent the slope variability for the four cases and among the four slope categories.

Table 6 summarizes the algorithm's performance for the four cases, in terms of $\overline{\omega}$ values. Although the four selected urban areas have different characteristics, overall, the derived BUSNs match the real BUSNs with acceptable accuracy, i.e., with $\overline{\omega}$ values above 60% in all of them. Note that the satisfactory validation of the derived BUSN from the algorithm verifies the analogy assumption we make earlier (in Section 2.2.1), i.e., the analogy between the BUSN and the street network's topology.

As is evident from Table 6, the average coverage percentage and slope variability follow the same trend, i.e., the highest coverage percentage corresponds to the lowest slope variability.

**Table 6.** Summary of the model performance for the four cases

|  | Total Real BUSN Length (km) | DACI (%) | $\overline{\omega}$ (%) | $\overline{\gamma}$ (%) |
|---|---|---|---|---|
| Los Angeles | 1,612 | 0.9 | 75.5 | 4.5 |
| Houston | 6,352 | 0.8 | 72.7 | 4.1 |
| Seattle | 1,857 | 0.9 | 65.6 | 3.9 |
| Baltimore | 2,461 | 0.9 | 59.0 | 2.6 |

## 4   Summary and Discussions

This study presents a novel algorithm for estimating below-ground urban stormwater networks based on Graph theory concepts and publicly available information. Most of the procedure is automatic, except for one empirical parameter that is user-
440 specified. Inputs of the algorithm are mostly land surface data, such as street network, topography, land use/land cover, and building footprints, that are readily available to the public and cover at least the whole U.S. We successfully validated the topology of the derived BUSNs at four U.S. cities on both west and east coasts, with the average coverage percentage varied in 59-76%.

Although we developed our proposed framework based on the publicly available datasets and designs manual in the U.S.,
it is flexible and can be adapted to other regions with different design criteria and data availability. Moreover, despite relying only on publicly available datasets that are not the most accurate available datasets, the model had satisfactory performance.

There are a few directions to further improve the algorithm, including but not limited to:

1. The quality and availability of input data for the algorithm can be further enhanced at the regional or larger scales, e.g., the street network data.

2. The DACI threshold in deriving BUSNs is an empirical, user-specified parameter in this study. Estimating it a priori based on the hydroclimatic conditions for any urban watershed can be achieved via a rigorous hydraulic analysis involving estimating peak runoff and adequately detailed BUSN hydraulic modeling.

3. We may generalize this DACI threshold parameter at the regional or larger scales based on the regional hydroclimate conditions (e.g., intensity-duration-frequency of extreme rainfall and peak runoff). These improvements are nevertheless
beyond the scope of this study and left for future work.

4. We may further expand our algorithm to account for drainage catchments in urban areas and break down the derived BUSNs into several subnetworks that follow the catchments.

5. The BUSN algorithm in this study is designed for separate sewer systems. Considering that combined sewer systems have different design criteria, the applicability of the algorithm for such systems requires further research.

Ultimately, our proposed algorithm for estimating BUSNs is a valuable tool to support the parameterization of large-scale urban hydrologic modeling, particularly in the areas where BUSN data are not available. It may also provide decision support in regional-scale urban planning from the angle of stormwater and flood management.

*Code and data availability.* The source code and the data generated from this study are available from the corresponding author upon reasonable request.

*Author contributions.* TC and HYL conceived the idea. TC designed and implemented the algorithm, and performed the analyses with inputs from HYL. Both authors contributed to the writing

*Competing interests.* H.-Y. Li acknowledges his financial interest in Pythias Analytics regarding the support from Sloan Foundation.

*Acknowledgements.* T. Chegini was supported by the University of Houston's internal funds and the Sloan Foundation via the Houston Advanced Research Center (contract no. UH0421). H.-Y. Li was also supported by the U.S. Department of Energy Office of Science Bio-
470 logical and Environmental Research as part of the Earth System Model Development program area through the collaborative, multi-program Integrated Coastal Modeling (ICoM) project.

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
