# Peer review of "An Algorithm for Deriving the Topology of Below-ground Urban Stormwater Networks"

_EGUsphere, 2022_

## Author Comment (AC1)

**Authors Response (AR) to RC #1**

*This paper presents a novel algorithm for deriving below-ground urban stormwater networks using graph theory concepts. The paper is well written, well-articulated, and presents application to both urban hydrologic modeling and broad Earth system modeling. Although the manuscript is heavy on the method description, the applicability of the algorithm is clearly illustrated with 4 case studies. I think some shifting of paragraphs are needed to make the paper flow better. I also have some minor comments for the authors to consider.*

**AR:** We would like to thank the reviewer for the constructive comments and criticism. Below are our point-to-point responses to the comments.

1. *It might be helpful to explain what "edge" and "node" mean at the beginning. It was clear later that edge means pipe and node means users but providing some contexts at the beginning would be helpful.*

   **AR:** We will add the explanations of them at the beginning in the revised manuscript.

2. *On page 4, at the beginning, I was confused about line 92-94. It seems counter-intuitive that higher weight results lower weighted BC value. Also, "a higher weight suggests a larger resistance to water flow and thus a lower flow rate." This part also confuses me. If the pipe is made out of the same materials, how will a higher weight lead to larger resistance to water flow? Is it because the length of the pipe is longer? If that's the case, then the pipe should be included because of its importance, right? I think the weighting process needs to be better explained up front to avoid confusion.*

   **AR:** We agree that the "weight" concept in Graph theory is somewhat counter-intuitive. However, we note that for computing BC we are looking for the shortest paths, i.e., minimizing the sum of edge weights. Therefore, we need to define the properties of interest for assigning edge weights, accordingly. For example, in the context of BUSN properties, our goal is to assign a higher BC value to a pipe with a higher flow capacity. Additionally, we know that some pipe properties, such as diameter, have a direct relationship with flow capacity and others, such as roughness, have an inverse relationship. Therefore, to minimize the edge weights, we transform those properties with a direct relationship in such a way that higher values correspond to lower weights. In contrast, we should transform those properties with an inverse relationship so that higher values correspond to higher weights. We will rephrase our usage of the weight concept in BUSN to clearly reflect this direct/inverse relationship.

3. *On page 8, bullet point 8, "those components that are unreachable after converting the network to an undirected graph by ignoring edge directions", did you check if these pipes are actually not important edges? How rigorous is this approach?*

   **AR:** By unreachable we meant those isolated group of edges that form relatively small subgraphs that are not connected to the other parts. So, the approach is rigorous since we are only eliminating those isolated subgraphs that are very small in comparison to the

whole network. The key sentence in bullet point 8 is "Then, we find the number of streets for each subnetwork and remove those subnetworks whose number of streets is less than the average street count of the subnetworks."

4. *I got a lot of questions when I saw Figure 5. For example, I wonder how pipes sizes are assigned based on BC and permissible min/max diameter. I found out that these were explained later in the manuscript. I think some indications directing the readers to the section where these are explained would be helpful.*

   **AR:** We agree with this comment and will make necessary rearrangement in the revised manuscript, e.g., moving the Fig. 5 to the end of section 2 after all the detailed explanation.

5. *On page 11, line 204, "assigning two lanes to those road types not listed in Table 2", why?*

   **AR:** The road types that are not listed in the table are generally for local-access and are small percentage of the total number of roads. They are usually two-way roads, so we considered them as two-lane roads. We will add this explanation in the revised manuscript.

6. *On page 12, line 216, maximum discharge for a pipe when depth of water is flowing at 94% of diameter, not full!*

   **AR:** We agree that the actual flow capacity of a pipe is when the water depth is at about 94% the maximum height that leads to about 8% difference between the actual and nominal hydraulic capacities. We will make note of this in the revised manuscript. Essentially, in our algorithm we compare the hydraulic capacity of pipes to assign them weights and compute their "relative significance". So, since we are using the full pipe diameter to compare the capacity of pipes, using full or 94% does not make a difference in their "relative significance".

7. *In table 3, what does the LULC number mean?*

   **AR:** By LULC we were referring to the land cover class. We will spell it out in the table.

8. *On page 20, in table 6, the last column appeared for the first time without giving any context. It was explained later in the manuscript, but some explanations are needed when it first appeared.*

   **AR:** Thanks for catching this. We will move the table to the end of Section 3 to make sure all explanations are provided beforehand.

9. *On page 25, I don't see any drainage pipes captured for the center of the city. Why is that the case?*

**AR:** As we mentioned in Section 3.1, for example, when discussing Figure 9a, the publicly available BUSN data have poor quality and cover only a portion of the urban areas. For the Baltimore case (page 25) the existing BUSN is not publicly available in the city center. We are not sure exactly why they the local government did not make the data for the city center publicly available.

10. *The first sentence in Introduction does not flow well. Please revise.*

    **AR:** We will rewrite the sentence in the revised manuscript.

11. *Is Figure 1 an original creation or is it obtained from other sources? Please ensure that IP is not infringed.*

    **AR:** This figure is adapted from Town of Gilbert, AZ (2022) but with substantial modifications. Below please see the comparison between the original version and our version.

[Figure]

*Figure R1. Source image from* Town of Gilbert, AZ (2022)

[Figure]

*Figure R2. Our adapted version of Figure 1*

---

## Author Comment (AC2)

**Authors Response (AR) to RC #2**

*This article addresses a highly needing yet challenging problem, deriving the topology of urban drainage networks from land surface data. A novel algorithm was developed and when applying to four various urban areas the accuracy (60-75%) is acceptable, especially given the complexity of the problem and uncertainties of the input data. Specific comments are as follows:*

**AR:** We would like to thank you for your constructive comments and criticism. We will try our best to address the issues that you've raised in the next revision. Here is our point-to-point response to the comments.

1. *The term "Below-ground Urban Stormwater Networks (BUSNs)" seems created by authors? Why not more commonly used term, such as "Urban Drainage Networks"?*

   **AR:** Since "Urban Drainage Network" is a general term that includes both surface and subsurface components, we used BUSN to explicitly reflect the objective of the manuscript, i.e., estimating topology of the below-ground components (not surface components such as street inlets) of an urban stormwater (not sanitary/combined sewer) drainage network.

2. *Although not explicitly said, Figure 1 and line #23 seem indicating that authors focused on separate sewer systems (i.e., not combined sewer systems) and only stormwater drainage networks (i.e., not sewer networks)? Noting there are hundreds of cities in the US that have combined sewer systems, how well would this algorithm apply to those systems?*

   **AR:** Yes, that is correct, our study only concerns with stormwater network. We will explicitly mention that Municipal Separate Storm Sewer System (MS4) is the subject of this study in the next revision. Regarding extensibility of our algorithm to combined sewer networks, it requires further study, since the design criteria of combined sewer networks are different than stormwater networks'.

3. *Validation was performed using a metric for coverage as the goal seems to be deriving the "topology". I'm curious if authors considered and compared slope and size of pipes? How would slope and size be implemented in large-scale urban hydrologic modeling?*

   **AR:** Unfortunately, the slope and size of existing BUSN are not available publicly for the urban areas that we selected for this study. In our algorithm, however, we do provide an estimate for the pipe slopes and sizes (lines 237-251) but since real data are not available, we cannot estimate their accuracy.

4. *Line 15: "urban population will grow from half to more than two-thirds of the total population by 2050." I'd suggest to delete "from half", or add "from half by 2008".*

   **AR:** Thanks for the suggestion, we will make this change in the next revision.

5. *Line 24: "most urban modules in existing hydrological models..." provide references and/or give examples.*

    **AR:** We will add examples in the next revision.

6. *Line 159: "60% of a pipe length from the real BUSN is within this buffer zone, the pipe is considered "covered"." Did authors consider other values as the criteria? I'm curious how sensitive this criteria would be.*

    **AR:** Yes, we tested the model sensitivity to this threshold using threshold values ranging from 50-80% and the difference in the total coverage percentage was in the order of 1%. So, the model is not very sensitive and we opted for using 60%.

7. *Line 381 vs. line 9: 59-76% vs. 60-75%. Which one is correct?*

    **AR:** Thanks for catching this. The correct values are 59-76%. We will fix this in the next revision.

---

## Author Comment (AC3)

**Authors Response (AR) to RC #3**

*The manuscript entitled "An Algorithm for Deriving the Topology of Below-ground Urban Stormwater Networks" proposes a novel algorithm for estimating Below-ground Urban Stormwater Networks (BUSNs) from existing data based on the Graph theory concepts. The paper is interesting. However, the manuscript has some shortcomings which need to be improved prior to its publication. The recommendation is that the article needs Major Revisions before it can be considered for publication. The following suggestions must be revised:*

**AR:** We would like to thank you for your constructive comments and criticism, which we will carefully address in the revised manuscript. Below are our point-to-point responses to the comments.

1. *The abstract should be carefully rewritten as English expression needs improving and the structure is not as clear as the main part of the paper. The novel algorithm needs more explanation.*

   **AR:** We will rewrite the abstract and add more explanation about the algorithm in the abstract.

2. *Now the approximate computation method of drainage capacity for urban flood modeling is a common method in the area where the BUSN data are sparse, this should be mentioned in the introduction section.*

   **AR:** Thanks for suggestion, we will include this in the introduction.

3. *There are many drainage catchments in urban city, and the drainage pipe network is generally laid out according to the catchments. How to consider this in the algorithm?*

   **AR:** Our proposed algorithm, provides BUSN, one of the most critical inputs, that are needed for flood modeling. However, using the generated BUSN in an urban hydrological model requires some post-processing operations to account for the interactions between the derived BUSN and other sewershed elements, which is beyond the scope of the current study.

4. *The article only describes the pipes without mentioning the rainwater nodes and inlets, which also play a great role in the urban flooding process.*

   **AR:** As the title of our manuscript suggests, our algorithm provides an estimation only for the below-ground elements of an urban stormwater network. The surface urban drainage elements such as street inlets and manholes are not the subject of this study. Deriving the spatial layout and distribution of those above-ground urban drainage components requires considering hydrologic characteristics of urban areas such as precipitation and the location of river network and other bodies. This is however outside the scope of this manuscript and left for a future study. We will add this to the final section as a future direction for the manuscript.

5. *Validation section is weakly written. It is verified by the "covered" of the distribution of the pipe network, which is relatively rough, and there is no comparison of key parameters such as pipe diameter, slope, and flow direction.*

   **AR:** We will improve the writing of the validation section in the revised manuscript. The current validation strategy is already the best we can come out with due to the availability and quality of real BUSN data over multiple urban areas. The validation strategy (even the algorithm itself) can be potentially further enhanced by adding more details for some small urban areas where BUSN data are available with good quality (which are very rarely available to the public at the first place). However, we may then lose the generality hence applicability over the regional scales, which is the utmost objective of this study. As mentioned in Section 3.1, unfortunately, the existing, publicly available BUSN data are very sparse. Furthermore, those data that are publicly available, generally, do not include pipe slope, size, and flow direction. As a result, we intentionally limited the scope of our study to only derive the components that we can validate, i.e., the topology of BUSNs not pipe sizes and slopes. Although we provide (hydraulicly feasible) estimates of size, slope, and flow direction for derived BUSNs, since we cannot validate them, we do not emphasize them as important products of our algorithm.

6. *The author should check the whole manuscript carefully, there are some errors in the interpretation of the diagrams.*

   **AR:** We will revise the interpretation, and the diagrams if needed, to make sure they are consistent with each other.

---

## Author Response (AR1)

Dear editor and reviewers,

The authors would like to thank the reviewers for your time and constructive comments. We believe that we have successfully addressed the reviewers' comments. Our point-to-point responses are listed below, where our responses are in black color and the reviewers' comments are in gray color and italic. **Please note that the line numbers we provide in the below are corresponding to the revised manuscript with the changes tracked.**

**Authors Response (AR) to RC #1**

*This paper presents a novel algorithm for deriving below-ground urban stormwater networks using graph theory concepts. The paper is well written, well-articulated, and presents application to both urban hydrologic modeling and broad Earth system modeling. Although the manuscript is heavy on the method description, the applicability of the algorithm is clearly illustrated with 4 case studies. I think some shifting of paragraphs are needed to make the paper flow better. I also have some minor comments for the authors to consider.*

**AR:** We would like to thank the reviewer for the constructive comments and criticism. Below are our point-to-point responses to the comments.

1. *It might be helpful to explain what "edge" and "node" mean at the beginning. It was clear later that edge means pipe and node means users but providing some contexts at the beginning would be helpful.*

   **AR:** We added an explanation for node and edges in **Lines 111-112 and Line 125**.

2. *On page 4, at the beginning, I was confused about line 92-94. It seems counter-intuitive that higher weight results lower weighted BC value. Also, "a higher weight suggests a larger resistance to water flow and thus a lower flow rate." This part also confuses me. If the pipe is made out of the same materials, how will a higher weight lead to larger resistance to water flow? Is it because the length of the pipe is longer? If that's the case, then the pipe should be included because of its importance, right? I think the weighting process needs to be better explained up front to avoid confusion.*

   **AR:** We agree that the "weight" concept in Graph theory is somewhat counter-intuitive, and we have tried our best to better introduce and use this concept. Note that for computing BC we are looking for the shortest paths, i.e., the higher the BC value, the lower the corresponding sum of the edge weights. Therefore, we need to define the properties of interest for assigning edge weights, accordingly. For example, in the context of BUSN properties, our goal is to assign a higher BC value to a pipe with a higher flow capacity. Additionally, we know that some pipe properties, such as diameter, have a direct relationship with flow capacity and others, such as roughness, have an inverse relationship. Therefore, to minimize the edge weights, we transform those properties with a direct relationship in such a way that higher/lower values correspond to lower/higher

weights, respectively. We rewrote Section 2.1.1 (**Lines 70-165**) to clarify and provide more context for the "weight" concept.

3. *On page 8, bullet point 8, "those components that are unreachable after converting the network to an undirected graph by ignoring edge directions", did you check if these pipes are actually not important edges? How rigorous is this approach?*

   **AR:** By unreachable we meant those isolated group of edges that form relatively small subgraphs that are not connected to the other parts, hence not important. So, the approach is rigorous since we are only eliminating those isolated subgraphs that are very small in comparison to the whole network. The key sentence in bullet point 8 is "Then, we find the number of streets for each subnetwork and remove those subnetworks whose number of streets is less than the average street count of the subnetworks."

4. *I got a lot of questions when I saw Figure 5. For example, I wonder how pipes sizes are assigned based on BC and permissible min/max diameter. I found out that these were explained later in the manuscript. I think some indications directing the readers to the section where these are explained would be helpful.*

   **AR:** We agree with this comment. We moved the figure to the end of Section 2 (**Page 18**) and its figure number changed to 6 from 5.

5. *On page 11, line 204, "assigning two lanes to those road types not listed in Table 2", why?*

   **AR:** The road types that are not listed in the table are generally for local-access and are small percentage of the total number of roads. They are usually two-way roads, so we considered them as two-lane roads. We added a new sentence in **Lines 260-261** to address this comment.

6. *On page 12, line 216, maximum discharge for a pipe when depth of water is flowing at 94% of diameter, not full!*

   **AR:** We agree that the actual flow capacity of a pipe is when the water depth is at about 94% the maximum height that leads to about 8% difference between the actual and nominal hydraulic capacities. Essentially, in our algorithm we compare the hydraulic capacity of pipes to assign them weights and compute their "relative significance". So, since we are using the full pipe diameter to compare the capacity of pipes, using full or 94% does not make a difference in their "relative significance". We add a sentence in **Lines 291-293** to explicitly mention this point.

7. *In table 3, what does the LULC number mean?*

   **AR:** By LULC we were referring to the land cover class. We changed LULC to Land Cover Type in Table 3 (**Page 14, Table 3**).

8. *On page 20, in table 6, the last column appeared for the first time without giving any context. It was explained later in the manuscript, but some explanations are needed when it first appeared.*

**AR:** Thanks for catching this. We moved Table 6 to **Page 29** and its corresponding paragraph to **Lines 430-433**.

9. *On page 25, I don't see any drainage pipes captured for the center of the city. Why is that the case?*

**AR:** As we mentioned in Section 3.1, for example, when discussing Figure 9a, the publicly available BUSN data have poor quality and cover only a portion of the urban areas. For the Baltimore case (page 25) the existing BUSN is not publicly available in the city center. We are not sure exactly why they the local government did not make the data for the city center publicly available. In the captions of Figures 12-15, we included a sentence that refer to the unavailability of public real BUSN data: "The white background pixels are grid cells that do not contain street and/or existing BUSN elements".

10. *The first sentence in Introduction does not flow well. Please revise.*

**AR:** Revised (**Line 13**).

11. *Is Figure 1 an original creation or is it obtained from other sources? Please ensure that IP is not infringed.*

**AR:** This figure is adapted from Town of Gilbert, AZ (2022) but with substantial modifications. We added reference to the original figure to Figure 1's caption (**Page 3**) Below in Figure R1 please see the comparison between the original version and our version.

[Figure]

*Figure R1a.. Source image from* *Town of Gilbert, AZ (2022)*

[Figure]

*Figure R1b. Our adapted version of Figure 1*

**Authors Response (AR) to RC #2**

*This article addresses a highly needing yet challenging problem, deriving the topology of urban drainage networks from land surface data. A novel algorithm was developed and when applying to four various urban areas the accuracy (60-75%) is acceptable, especially given the complexity of the problem and uncertainties of the input data. Specific comments are as follows:*

**AR:** We would like to thank the reviewer for the constructive comments. We have carefully addressed them. Here are our point-to-point responses.

1. *The term "Below-ground Urban Stormwater Networks (BUSNs)" seems created by authors? Why not more commonly used term, such as "Urban Drainage Networks"?*

   **AR:** Since "Urban Drainage Network" is a general term that includes both surface and subsurface components, we used BUSN to explicitly reflect the objective of the manuscript, i.e., estimating topology of the below-ground components (not surface components such as street inlets) of an urban stormwater (not sanitary/combined sewer) drainage network.

2. *Although not explicitly said, Figure 1 and line #23 seem indicating that authors focused on separate sewer systems (i.e., not combined sewer systems) and only stormwater drainage networks (i.e., not sewer networks)? Noting there are hundreds of cities in the US that have combined sewer systems, how well would this algorithm apply to those systems?*

   **AR:** Yes, that is correct, our study only concerns with stormwater networks in separate sewer systems, which are dominant in the U.S. and China. In the Introduction section, we added a new paragraph (**Lines 21-29**) and modified the two subsequent paragraphs (**Lines 30-47**) to explicitly mention that below-ground elements of Municipal Separate Storm Sewer System (MS4) is the subject of this study. Our algorithm can't be directly applied to combined sewer systems, since the design criteria of combined sewer networks

are different than stormwater networks. We added a note about this limitation in the Discussion section (**Lines 457-458**).

3. *Validation was performed using a metric for coverage as the goal seems to be deriving the "topology". I'm curious if authors considered and compared slope and size of pipes? How would slope and size be implemented in large-scale urban hydrologic modeling?*

   **AR:** Unfortunately, the slope and size of existing BUSN are not available publicly for the urban areas that we selected for this study. In our algorithm, however, we do provide an estimate for the pipe slopes and sizes (see Lines 247-261) but since real data are not available, we cannot directly estimate their accuracy. We might be able to indirectly validate our estimated BUSN slope and size values when applying them in large-scale urban hydrologic modeling, e.g., by examining how well the model captures the observed urban streamflow when using the estimated BUSN slope and size values. This indirect validation requires lots of additional work and is left for future study.

4. *Line 15: "urban population will grow from half to more than two-thirds of the total population by 2050." I'd suggest to delete "from half", or add "from half by 2008".*

   **AR:** Thanks for the suggestion, we deleted "from half" in **Line 15**.

5. *Line 24: "most urban modules in existing hydrological models..." provide references and/or give examples.*

   **AR:** We added a few references in **Lines 36-42**.

6. *Line 159: "60% of a pipe length from the real BUSN is within this buffer zone, the pipe is considered "covered"." Did authors consider other values as the criteria? I'm curious how sensitive this criteria would be.*

   **AR:** Yes, we tested the model sensitivity to this threshold using threshold values ranging from 50-80% and the difference in the total coverage percentage was in the order of 1%. So, the model is not very sensitive and we opted for using 60%. Below Figure R2 shows a comparison of the coverage percentage for the four case studies and with different threshold values between 50 and 80 percent with a step size of 5%. We added a note on this on **Lines 214-215**.

[Figure]

Figure R2: sensitivity analysis of the threshold value.

7. *Line 381 vs. line 9: 59-76% vs. 60-75%. Which one is correct?*

**AR:** Thanks for catching this. The correct values are 59-76%. We change the value in the abstract (**Line 9**).

**Authors Response (AR) to RC #3**

*The manuscript entitled "An Algorithm for Deriving the Topology of Below-ground Urban Stormwater Networks" proposes a novel algorithm for estimating Below-ground Urban Stormwater Networks (BUSNs) from existing data based on the Graph theory concepts. The paper is interesting. However, the manuscript has some shortcomings which need to be improved prior to its publication. The recommendation is that the article needs Major Revisions before it can be considered for publication. The following suggestions must be revised:*

**AR:** We would like to thank the reviewer for the constructive comments and criticism, which we have carefully addressed in the revised manuscript. Below are our point-to-point responses to the comments.

1. *The abstract should be carefully rewritten as English expression needs improving and the structure is not as clear as the main part of the paper. The novel algorithm needs more explanation.*

   **AR:** We modified the abstract to make it smoother and clearer.

2. *Now the approximate computation method of drainage capacity for urban flood modeling is a common method in the area where the BUSN data are sparse, this should be mentioned in the introduction section.*

   **AR:** Thanks for suggestion, we cited several additional references on this method in **Lines 36-42**.

3. *There are many drainage catchments in urban city, and the drainage pipe network is generally laid out according to the catchments. How to consider this in the algorithm?*

   **AR:** Drainage catchments, also noted as sewersheds in urban hydrology, are to collect excess rainfall and discharge them into BUSN or river networks. In general, sewersheds are indeed closely related to BUSN's transport capacity since more excess rainfall requires larger hydraulic transport capacity of BUSN, which is controlled by the sizes and slopes of BUSN pipes. In our study, the focus is on BUSN's topology only, which is at most indirectly related to BUSN's hydraulic transport capacity. Therefore, explicit consideration of drainage catchments is beyond the scope of current study and left for future work. We have added this consideration of drainage catchments in the end of the manuscript as one of the future directions. See **Lines 455-456**.

4. *The article only describes the pipes without mentioning the rainwater nodes and inlets, which also play a great role in the urban flooding process.*

   **AR:** As the title of our manuscript suggests, our algorithm provides an estimation only for the below-ground elements of an urban stormwater network. The surface urban drainage elements such as street inlets and manholes are not the subject of this study. Deriving the spatial layout and distribution of those above-ground urban drainage

components require considering hydrologic characteristics of urban areas such as precipitation and the location of river network and other bodies. This is however outside the scope of this manuscript and left for a future study. We used the term Municipal Separate Storm Sewer System (MS4) in the Introduction (**Lines 21-47**) to make it clearer that this study only concerns with the below-ground elements of MS4s not surface elements such as street inlets and rainwater nodes.

5. *Validation section is weakly written. It is verified by the "covered" of the distribution of the pipe network, which is relatively rough, and there is no comparison of key parameters such as pipe diameter, slope, and flow direction.*

    **AR:** Based on the existing validation data, the current validation strategy is already the best we can propose that is applicable to a wide range of places. The validation strategy (even the algorithm itself) can be potentially further enhanced by adding more details for some small urban areas where the BUSN data are available with good quality (which are very rarely available to the public at the first place). However, we may then lose the generality hence applicability over the regional scale, which is the utmost objective of this study. As mentioned in Section 3.1, unfortunately, the existing, publicly available BUSN data are very sparse and even those data that are publicly available, generally, do not include pipe slope, size, and flow direction. As a result, we intentionally limited the scope of our study to only derive the product that we can validate, i.e., the topology of BUSNs, but not pipe sizes and slopes. Although we provide (hydraulicly feasible) estimates of size, slope, and flow direction for derived BUSNs, we cannot validate them, we thus do not consider them as a product of our algorithm. We added a discussion on this point in **Lines 208-210**.

6. *The author should check the whole manuscript carefully, there are some errors in the interpretation of the diagrams.*

    **AR:** We modified Figure 3 (**Page 9**) and Figure 6 (**Page 18**) to make sure that they match the description provided in the text. With that, the associated interpretation is now adequate.